# Equivariance by Contrast: Identifiable Equivariant Embeddings from Unlabeled Finite Group Actions

**Tobias Schmidt**[1,2]**, Steffen Schneider**[1,2,3*] **and Matthias Bethge**[3*]
[1]Institute of Computational Biology, Helmholtz Munich
[2]Munich Center for Machine Learning (MCML)
[3]Tübingen AI Center

## Abstract

We propose Equivariance by Contrast (EbC) to learn equivariant embeddings from observation pairs $(\mathbf{y}, g \cdot \mathbf{y})$, where $g$ is drawn from a finite group acting on the data. Our method jointly learns a latent space and a group representation in which group actions correspond to invertible linear maps—without relying on group-specific inductive biases. We validate our approach on the infinite dSprites dataset with structured transformations defined by the finite group $G := (R_m \times \mathbb{Z}_n \times \mathbb{Z}_n)$, combining discrete rotations and periodic translations. The resulting embeddings exhibit high-fidelity equivariance, with group operations faithfully reproduced in latent space. On synthetic data, we further validate the approach on the non-abelian orthogonal group $O(n)$ and the general linear group $GL(n)$. We also provide a theoretical proof for identifiability. While broad evaluation across diverse group types on real-world data remains future work, our results constitute the first successful demonstration of general-purpose encoder-only equivariant learning from group action observations alone, including non-trivial non-abelian groups and a product group motivated by modeling affine equivariances in computer vision.

## 1 Introduction

In many real-world inference problems, the relationship between observations is governed by structured transformations. The same sample may be observed before and after an "action" has been applied. In computer vision, an object may be observed in the form of an image before and after rotations, translations, or other types of transformations have been applied [6, 11, 32]. In biology, large-scale single-cell transcriptomic datasets increasingly contain observations of cells before and after perturbations in the form of gene knockouts [17] or pharmacological intervention [41]. In neuroscience, neural activity reflects changing brain states under sensory input or behavioral output [45]. In all these cases, the key to understanding the data lies not only in modeling individual observations but also in capturing the structured relationships between them.

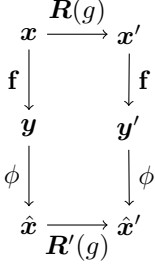

Figure 1: Commutative diagram showing data and model for EbC.

This motivates the goal of learning *equivariant* embeddings. In this embedding space, actions are represented by linear transformations. To address this challenge in a theoretically grounded way, we adopt a perspective rooted in nonlinear Independent Component Analysis (ICA) and group theory. Suppose that each observation $\boldsymbol{y} \in Y$ arises from a latent representation $\boldsymbol{x} \in X$ through an unknown, injective nonlinear mixing function $\mathbf{f}$, i.e., $\boldsymbol{y} = \mathbf{f}(\boldsymbol{x})$. Nonlinear ICA aims to invert this process: to learn an encoder $\phi \approx \boldsymbol{f}^{-1}$ that recovers the latent structure from the observed data.

---

*Co-corresponding authors: steffen.schneider@helmholtz-munich.de, matthias.bethge@bethgelab.org.

39th Conference on Neural Information Processing Systems (NeurIPS 2025).

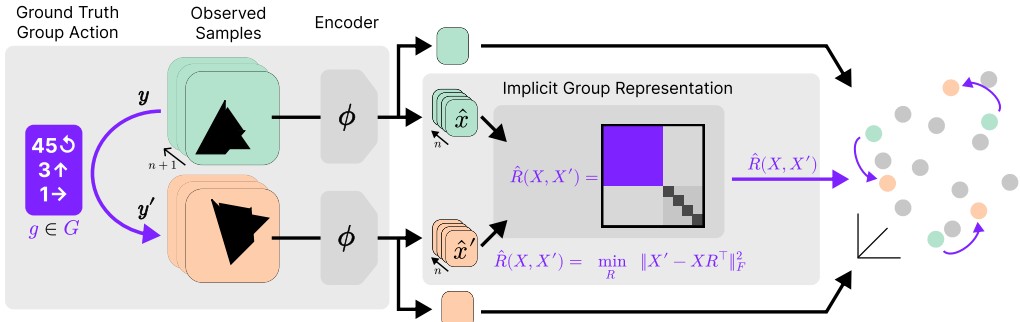

Figure 2: **Overview of the approach**. Left to right: One batch of observed sample consists of $n + 1$ paired samples $\{(\boldsymbol{y}_i, \boldsymbol{y}'_i)\}_{i=0}^{n+1}$ where $\boldsymbol{y}'_i$ are related to $\boldsymbol{y}_i$ via an unkown group action $g$ such that $\boldsymbol{y}'_i = g \cdot \boldsymbol{y}_i$. An encoder $\phi$ maps these observations into latent space $\{(\hat{\boldsymbol{x}}_i, \hat{\boldsymbol{x}}'_i)\}_{i=0}^{n+1}$, where $n$ samples are used to estimate a representation $\hat{\boldsymbol{R}}$ of the group.

The nonlinear ICA problem becomes tractable by additional assumptions about the structure of the data-generating process [18, 19, 21, 22, 56]. Here, we leverage the property that many datasets do not consist of isolated samples but of pairs $(\boldsymbol{y}, \boldsymbol{y}')$, where $\boldsymbol{y}'$ is a transformed version of $\boldsymbol{y}$. Group theory provides a formalism to describe these transformations as the elements of a group $G$.

Each group element $g \in G$ can act on a latent $\boldsymbol{x} \in X$, yielding a transformed latent $g\boldsymbol{x}$. This operation is known as a *group action*. In observation space, we introduce the shorthand $\boldsymbol{y}' := g \cdot \boldsymbol{y}$ to denote the respective relation between $\boldsymbol{y}' = \mathbf{f}(g\boldsymbol{x})$ and $\boldsymbol{y} = \mathbf{f}(\boldsymbol{x})$. In the latent space, these group actions have a particularly simple form: they correspond to linear maps. Formally, a *group representation* is a homomorphism $\boldsymbol{R} : G \mapsto GL(X)$ which maps each group element to an invertible matrix acting on the vector space $X$. Thus, while the transformation $g \cdot \boldsymbol{y}$ may have complicated non-linear effects in observation space, in a suitable latent space it can be reduced to a linear relationship of the form:

$$\boldsymbol{x}' = g\boldsymbol{x} = \boldsymbol{R}(g)\boldsymbol{x}. \tag{1}$$

Our goal is to infer this relation directly from pairs $(\boldsymbol{y}, g \cdot \boldsymbol{y})$ of observable data and learn an encoder $\phi : Y \mapsto X'$ and a representation $\boldsymbol{R}'$ of the group such that

$$\phi(g \cdot \boldsymbol{y}) = \boldsymbol{R}'(g)\phi(\boldsymbol{y}), \tag{2}$$

and $\boldsymbol{R}' : G \mapsto GL(X')$ is a representation of $G$ (potentially different from $\boldsymbol{R}$) on the vector space $X'$. Learning such representations directly from data is difficult. A growing body of work has approached this problem by leveraging pairs $(\boldsymbol{y}, g \cdot \boldsymbol{y})$ even when the specific group element $g$ is unknown [15, 28, 52], with varying constraints: CARE [15] restricts itself to orthogonal representations on the hypersphere, STL [52] allows nonlinear equivariant relations in latent space, and the neural fourier transform [NFT; 28] requires to learn a generative model of the data.

Here, we propose *Equivariance by Contrast (EbC)*, to the best of our knowledge the first *encoder-only* method that learns *general linear* group representations from group action observations with a formal identifiability guarantee. In § 2, we present the algorithm which jointly learns the encoder $\phi$ and an implicit group representation via contrastive learning, without generative modeling or group-specific architectural biases. In § 3, we show that EbC recovers the true latent space and the underlying group representation up to a linear transformation. In §§ 4–6, we evaluate EbC on synthetic and structured vision datasets, including finite product groups $G := (R_m \times \mathbb{Z}_n \times \mathbb{Z}_n)$ and non-abelian groups such as $O(n)$ and $GL(n)$. EbC achieves high-fidelity equivariant embeddings across diverse settings.

## 2 Learning group-equivariant representations with contrastive learning

Figure 2 outlines our approach: Similar to previous work [15, 28, 36, 52], we assume that data come in the form of batches, which are grouped into $n + 1$ pairs undergoing the same action $g$. This form of data is common in various scientific fields, for example, in neuroscience and biology. Pairs can also be derived from time-series data under the assumption that nearby points are governed by a shared action ["slowness prior"; 26].

The intuition behind our objective function is depicted in Figure 3: The model has access to a set of mixed samples related via the group action ($\boldsymbol{Y}$ and $\boldsymbol{Y}'$), along with a query sample $\boldsymbol{y}$. The objective

is to infer the group action from the examples $\boldsymbol{Y}$ and $\boldsymbol{Y}'$, apply it to the query, and select the correct answer $\boldsymbol{y}'$ among a set of options that include the correct answer alongside negative samples $\boldsymbol{y}'' \in S$. In our case, the set $S$ contains negative samples randomly selected from the dataset, which cover all potential mismatches (different content, different group action, etc.) alongside the positive sample. We use contrastive learning to encode this objective in the likelihood

$$p_\phi(\boldsymbol{y}' \mid \boldsymbol{y}, \boldsymbol{Y}, \boldsymbol{Y}', S) = \frac{\exp\left(-\|\boldsymbol{u}_\phi(\boldsymbol{y}, \boldsymbol{Y}, \boldsymbol{Y}') - \phi(\boldsymbol{y}')\|^2\right)}{\sum_{\boldsymbol{y}'' \in S} \exp\left(-\|\boldsymbol{u}_\phi(\boldsymbol{y}, \boldsymbol{Y}, \boldsymbol{Y}') - \phi(\boldsymbol{y}'')\|^2\right)}. \tag{3}$$

The shorthand $\boldsymbol{u}_\phi$ denotes the operation of inferring the linear representation of the group element, $\hat{\boldsymbol{R}}(\phi(\boldsymbol{Y}), \phi(\boldsymbol{Y}'))$, and then applying it to the feature vector of the reference sample $\phi(\boldsymbol{y})$:

$$\boldsymbol{u}_\phi(\boldsymbol{y}, \boldsymbol{Y}, \boldsymbol{Y}') = \hat{\boldsymbol{R}}(\phi(\boldsymbol{Y}), \phi(\boldsymbol{Y}'))\phi(\boldsymbol{y}) \tag{4}$$

$$\hat{\boldsymbol{R}}(\boldsymbol{X}, \boldsymbol{X}') = \min_{\boldsymbol{R} \in \mathrm{GL}(d)} \|\boldsymbol{X}' - \boldsymbol{X}\boldsymbol{R}^\top\|_F^2. \tag{5}$$

To find the optimal feature encoder $\phi$, we optimize the likelihood across all pairs of samples and uniformly sampled negative examples,

$$\min_\phi \mathcal{L}[\phi] = -\mathbb{E}_{\boldsymbol{y}, \boldsymbol{y}', \boldsymbol{Y}, \boldsymbol{Y}', S}\left[\log p_\phi(\boldsymbol{y}' \mid \boldsymbol{y}, \boldsymbol{Y}, \boldsymbol{Y}', S)\right], \tag{6}$$

which is related to the InfoNCE loss [37] with the additional structure required for group learning.

**Separating content and style** In many cases, we want to produce embedding spaces in which we can separate *what* is transformed from *how* it is transformed. Intuitively, we consider these aspects of the latent representation to encode *content* and *style*. The content is the part of the representation that is expected to be invariant w.r.t. the group action $g$, whereas the style is expected to be equivariant w.r.t. the group action. In practice, it is therefore useful to split the vector space induced by $\phi$ into an equivariant and invariant part. Algorithmically, this is done by imposing additional structure on the matrix in Eq. (5), constraining the minimization across matrices of the form $\mathrm{diag}(GL_n, I_m) \subset GL(m + n)$. The representation we learn then has the form

$$\hat{\boldsymbol{R}}_{n+m}(\boldsymbol{X}, \boldsymbol{X}')' = \begin{pmatrix} \hat{\boldsymbol{R}}_n(\boldsymbol{X}, \boldsymbol{X}') & \boldsymbol{0}_{n \times m} \\ \boldsymbol{0}_{m \times n} & \boldsymbol{I}_m \end{pmatrix}. \tag{7}$$

This results in an encoder $\phi$ with a $n$-dimensional equivariant subspace, and a $m$-dimensional invariant subspace. We discuss the theoretical properties of this parametrization below. Note the conceptual similarity to subspace contrastive learning discussed in [44]. The difference here is that we do not use multiple instances of the contrastive loss but a single contrastive loss that directly learns a structured feature space.

## 3 Equivariance by Contrast is identifiable

By formalizing assumptions about the data-generating process, we can derive identifiability guarantees for the algorithm described in the previous section.

**Dataset.** We define the dataset in terms of the underlying data-generating process: Let $\boldsymbol{x} \in V \subseteq \mathbb{R}^d$ be a vector describing the ground truth latent components, let $g$ be an element of a group $G$, and let $\mathbf{f} : V \to \mathbb{R}^D$ be an injective map. Assume that for each group element $g$, we have at least $M$ pairs of samples $(\boldsymbol{y}_j, \boldsymbol{y}_j')$ with

$$\boldsymbol{y}_i = \mathbf{f}(\boldsymbol{x}_i), \qquad \boldsymbol{y}_i' = \mathbf{f}(\boldsymbol{R}(g)\boldsymbol{x}_i) \qquad i = 1, \ldots, M \tag{8}$$

where $\boldsymbol{R} : G \to \mathrm{GL}(d, \mathbb{R})$ is a representation of $G$ on $V$. If $\boldsymbol{R}$ is structured accordingly (Eq. 7), $\boldsymbol{x}$ can be decomposed into an equivariant *style* and an invariant *content*, which is denoted as $\boldsymbol{c}$ further below in the experimental sections.

**Implicit group representations.** We model the group representation using the non-parametric approach outlined in Eq. 5. Assume that for each group element $g \in G$, we are given two matrices $\boldsymbol{X}, \boldsymbol{X}' \in \mathbb{R}^{M \times d}$, $M > d$, where the row vectors $(\boldsymbol{x}_i, \boldsymbol{x}_i')$ are related via the group element as $\boldsymbol{x}_i' = g\boldsymbol{x}_i$, $i \in \{1, \ldots, M\}$. As a shorthand, we write $\boldsymbol{X}' = g\boldsymbol{X}$. Then, the expression

$$\hat{\boldsymbol{R}}(\boldsymbol{X}, \boldsymbol{X}') = \min_{\boldsymbol{R} \in \mathrm{GL}(d)} \|\boldsymbol{X}' - \boldsymbol{X}\boldsymbol{R}^\top\|_F^2 \Leftrightarrow \hat{\boldsymbol{R}}(\boldsymbol{X}, \boldsymbol{X}') = (\boldsymbol{X}^\top\boldsymbol{X})^{-1}(\boldsymbol{X}^\top\boldsymbol{X}') \tag{9}$$

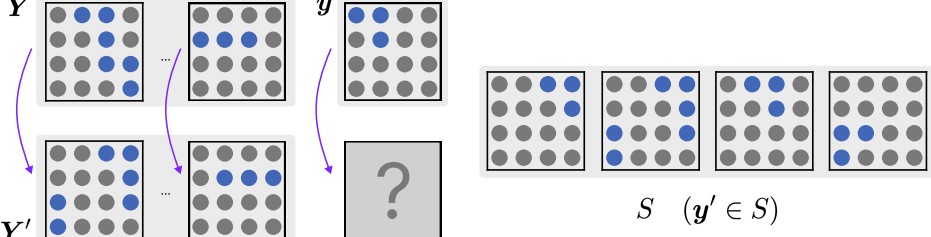

Figure 3: **Intuitive explanation of the training objective**. Given pairs of samples $(\boldsymbol{Y}, \boldsymbol{Y}')$ connected by an action $g$, infer the most likely action, and apply it to a novel sample $\boldsymbol{y}$ with different content. The training objective is contrastive, and aims to select the correct matching sample $\boldsymbol{y}'$ from a selection of positive/negative samples $S$.

is a representation of $G$ with $\boldsymbol{R}(g) = \hat{\boldsymbol{R}}(\boldsymbol{X}, g\boldsymbol{X})$ for each $g \in G$. In practice, we do not have access to $(\boldsymbol{X}, \boldsymbol{X}')$ directly, but to a nonlinear projection of these points via the mixing function $\mathbf{f}$, denoted as $(\mathbf{f}(\boldsymbol{X}), \mathbf{f}(\boldsymbol{X}')) = (\boldsymbol{Y}, \boldsymbol{Y}')$. We map these mixed samples to a feature space using a learnable encoder $\phi : \mathbb{R}^D \to \mathbb{R}^d$ and insert the resulting matrices into Eq. 9. Our goal is to optimize $\phi$ such that $\hat{\boldsymbol{R}}(\phi(\boldsymbol{Y}), \phi(\boldsymbol{Y}'))$ becomes a representation of $G$. An advantage of this approach is that both the feature space and the group representation are fully defined via the feature encoder $\phi$.

**Equivariance by Contrast learns group representations.** Our theory builds on the canonical discriminative form discussed by Roeder et al. [40], which requires conditions on the diversity of the dataset to be fulfilled. Specifically, the latents and feature spaces of $\phi$ need to "sufficiently vary" across the points in $S$. In addition, the vectors of differences $\phi(\boldsymbol{y}_A) - \phi(\boldsymbol{y}_B)$ with $\boldsymbol{y}_A, \boldsymbol{y}_B$ columns of $\boldsymbol{S}$ need to span a $d$-dimensional feature space. The conditions are discussed in detail in Appendix A. This gives the following main results for our model:

**Theorem 1** (Identitiability of the Group Representation; informal)**.** *Assume that $p_\phi = p_{\mathbf{f}^{-1}}$, and let the dataset satisfy the diversity conditions mentioned above. Define $\mathbf{h} := \phi \circ \mathbf{f}$. Then, for all points $\boldsymbol{x}$ in the support of the dataset:*

*(a) We recover the original vector space $\mathbf{h}(\boldsymbol{x}) = \boldsymbol{L}\boldsymbol{x}$, up to an ambiguity $\boldsymbol{L} \in \mathrm{GL}(d)$.*

*(b) We recover a representation of the group, $\hat{\boldsymbol{R}}(\mathbf{h}(\boldsymbol{X}), \mathbf{h}(g\boldsymbol{X})) = \boldsymbol{L}\boldsymbol{R}(g)\boldsymbol{L}^{-1}$.*

Proof sketch. *With a minor modification, Eq. 3 follows the canonical discriminative form for which Roeder et al. [40] proved linear identifiability. Due to the use of the more flexible Euclidean loss, we instead obtain affine identifiability for $\mathbf{h}(\boldsymbol{x}) = \boldsymbol{L}\boldsymbol{x} + \boldsymbol{b}$. The second condition requires $\hat{\boldsymbol{R}}\phi(\boldsymbol{x}) = \boldsymbol{L}\boldsymbol{R}\boldsymbol{x}$ and implies $\boldsymbol{b} = 0$. From these two conditions and the specific definition of our model, we can derive the results (a) and (b). The full proof is given in Appendix A.*

**Corollary 1** (Equivariance)**.** *We have $\boldsymbol{x} \in V$, $\boldsymbol{R}(g)$ is a representation of the group in $V$; we have $\mathbf{h} : V \to W$ where we define $\mathbf{h} := \phi \circ \mathbf{f}$, and $\boldsymbol{R}'(g) = \hat{\boldsymbol{R}}(\mathbf{h}(\boldsymbol{X}), \mathbf{h}(g\boldsymbol{X}))$ is the representation on $W$. Then, we have*

$$\mathbf{h}(g\boldsymbol{x}) = g\mathbf{h}(\boldsymbol{x}) \tag{10}$$

*Proof.* The result follows from Theorem 1. For all $\boldsymbol{x}$ we have $\mathbf{h}(g\boldsymbol{x}) = g\mathbf{h}(\boldsymbol{x})$. We insert the representation of $g$ on $V$ on the LHS, and on $W$ on the RHS, to obtain $\mathbf{h}(\boldsymbol{R}(g)\boldsymbol{x}) = \boldsymbol{R}'(g)\mathbf{h}(\boldsymbol{x})$. We insert Thm. 1a to obtain $\boldsymbol{L}\boldsymbol{R}(g)\boldsymbol{x} = \boldsymbol{R}'(g)\boldsymbol{L}\boldsymbol{x}$. From here we obtain $\boldsymbol{L}\boldsymbol{R}(g)\boldsymbol{L}^{-1}\boldsymbol{x} = \boldsymbol{R}'(g)\boldsymbol{x}$ which we showed in Thm. 1b. □

## 4 Experiment Setup

We validate our proposed model on a set of diverse datasets with different underlying groups.

**Synthetic Group Datasets** We first consider three types of synthetic datasets following the data-generating process in Eq. 8 matching the assumptions required for Theorem 1. Group elements are taken from a subgroup of special orthogonal group $SO(n)$, the orthogonal group $O(n)$ and the general linear group $GL(n)$. We start by sampling random group elements $g \sim p_G(g)$ from $G \in \{SO(n), O(n), GL(n)\}$. We sample at least $m$ (with $m > d$) random vectors $\boldsymbol{x} \in \mathbb{S}^n$ from the unit sphere and sample a content component $\boldsymbol{c} \in C$ from a finite set of vectors $C \subset S^d$. From

$(\boldsymbol{R}(g), \{\boldsymbol{x}_i\}^m, \boldsymbol{c})$ we generate $\boldsymbol{Y} = \{\boldsymbol{y}_i, \boldsymbol{y}_i'\}^m$, where $\boldsymbol{y}_i = \mathbf{f}(\boldsymbol{x}_i, \boldsymbol{c})$ and $\boldsymbol{y}_i' = \mathbf{f}(\boldsymbol{R}(g)\boldsymbol{x}_i, \boldsymbol{c})$. We repeat this sampling procedure until we reach the desired dataset size. The nonlinear injective mixing function $\mathbf{f}$ is parameterized by a random 3-layer MLP [18] followed by a random linear map to a 50-dimensional space. For the full dataset we sample a maximum of 1000 matrices $\boldsymbol{R}(g)$ and a total of 1M pairs $(\boldsymbol{y}, \boldsymbol{y}')$. Unless stated otherwise, we choose $n = 3$, $d = 3$, $|C| = 100$. In Appendix C.1 we show additional results for $n \in 3, 5, 7, 9$ and in Appendix C.2 we report results on data efficiency. According to the assumptions of Theorem 1, the relationship $\boldsymbol{R}(g)\boldsymbol{x}_i$ in the data-generating process does not include noise. But we show a variation of this experiment In Appendix C.3 that includes noise in the application of the group action.

**Infinite dSprites**  We leverage infinite dSprites [8, idSprites], an extension of dSprites [33] for validation on more diverse visual data. We subsample all images to a resolution of $64 \times 64$ The dataset allows for rich variations in the content and ensures that the resulting objects do not have any symmetries. By default, we use the same number of varying factors as in dSprites, namely 3 random shapes of 6 different sizes. For our purposes, we consider the combination of shape and size to be the content. Each of these objects may then be oriented in 40 different ways or have one of $32 \times 32$ x or y position translations. We consider the orientations and translations to be our groups of interest and sample the data such that we obtain closed groups. By default, we sample the dataset such that any paired sample $(\boldsymbol{y}, \boldsymbol{y}')$ undergoes a combination of these three types of transformations.

**Training Protocol**  We use a three layer MLP with 512 hidden units for $\phi$. We fit the implicit representation $\hat{\boldsymbol{R}}(\boldsymbol{Y}, \boldsymbol{Y}')$ using the `gels` least squares solver in PyTorch. By default, we use 84 sample pairs for idSprites and 12 for the synthetic data in $\boldsymbol{Y}$. Each batch has 1024 positive and 16k negative samples. We train the model for 20k with Adam [25] with learning rate $10^{-3}$. We perform 80/10/10 train/valid/test splits. Metrics are computed out-of-sample, by fitting on one part of the valid split, and reporting on the other. Unless stated otherwise, standard deviations and confidence intervals are reported across three dataset and three model seeds. More details in Appendix B.

**Baselines**  We leverage standard InfoNCE [37] training as a baseline (amounting to setting $\boldsymbol{R} = \boldsymbol{I}$), similar to SimCLR [2]. We also consider dynamics contrastive learning [30] using a linear dynamical system (LDS) or a switching linear dynamical system (SLDS). We use the same training protocol for the baselines as we do for our own EbC model (same encoder, batch size, training iterations, optimizer, and data loader).

**Metrics**  Our metrics aim to quantify the representation of the content, the quality of the group representation, and the linear identifiability of the latent space. To measure the quality of the representation of the content we fit a Logistic Regression $(\boldsymbol{W}, \boldsymbol{b})$ and predict the class label $j \in K$ of content $\boldsymbol{c}_j$:

$$\mathrm{Acc}(C, K) := \text{TopK-Acc}(\sigma(\boldsymbol{W}\mathbf{h}(\boldsymbol{x}) + \boldsymbol{b}) \mid \boldsymbol{Y}') \qquad \sigma(\cdot) := \mathrm{softmax}(\cdot) \tag{11}$$

Likewise, to quantify the linear identifiability of the embedding space (Theorem 1a), we measure:

$$R^2(x) := \max_{\boldsymbol{L} \in GL(n)} R_{\boldsymbol{x}}^2(\boldsymbol{L}\hat{\mathbf{h}}(\boldsymbol{x}), \boldsymbol{x}). \tag{12}$$

To measure the quality of the group representation we introduce two different metrics. First we introduce an $R^2$ based metric which we derive from Theorem 1b:

$$R^2(G) := R_{\boldsymbol{x}}^2[\boldsymbol{L}^*\hat{\boldsymbol{R}}\mathbf{h}(\boldsymbol{x}), \boldsymbol{R}\boldsymbol{x}], \quad \boldsymbol{L}^* = \arg\max_{\boldsymbol{L}} \|\boldsymbol{X} - h(\boldsymbol{X})\boldsymbol{L}^\top\|_F^2 \tag{13}$$

To measure the group representation without access to the ground truth latent representation $\boldsymbol{x}$, we introduce an Accuracy over the KNN lookup of the transformed input vector $\boldsymbol{y}'$:

$$\mathrm{Acc}(G, K) := \text{TopK-Acc}(\text{kNN}(\boldsymbol{Y}) \mid \boldsymbol{Y}') \qquad \text{kNN}(\boldsymbol{y}) := \arg\max_{\boldsymbol{j} \in \boldsymbol{J}} ||\phi(\boldsymbol{j}) - \hat{\boldsymbol{R}}\phi(\boldsymbol{y})||^2 \tag{14}$$

To compute this metric, we sample 20k random pairs $\boldsymbol{y}, \boldsymbol{y}'$ to solve a 20k-class classification problem. We compute the accuracy cross-validation across every of the 20k samples.

The metrics defined above can be split into two categories: $R^2(x)$ and $R^2(G)$ require access to the ground truth latents and hence can only be computed on synthetic toy datasets for which we have full knowledge about the data-generating process. In practice, this is not the case. Therefore, we defined a second group of metrics, including $\mathrm{Acc}(G, K)$ and $\mathrm{Acc}(C)$, that are applicable in the case where there is no access to the ground truth latents. We refer to appendix C.4 for a study on the relationship of $R^2(G)$ and $\mathrm{Acc}(G, K)$, which suggests we can use $\mathrm{Acc}(G, K)$ as a proxy of $R^2(G)$.

Table 1: Overview. We vary the group $G$ used for the data-generating process, and consider a non-linear mixing through a neural network ("non-linear") as well as the infinite dSprites dataset. We report the $R^2$ on the equivariant part of the embedding, and the Accuracy (in %) on the invariant part of the embedding for identifying the content information.

| Group (G) | $SO_3$ | | | $O_3$ | | | $GL_3$ | | | $R_m \times \mathbb{Z}_n \times \mathbb{Z}_n$ | |
|---|---|---|---|---|---|---|---|---|---|---|---|
| Obs. (**f**) | non-linear | | | non-linear | | | non-linear | | | indSprites | |
| Metric | $R^2(x)$ | $R^2(G)$ | Acc(C) | $R^2(x)$ | $R^2(G)$ | Acc(C) | $R^2(x)$ | $R^2(G)$ | Acc(C) | Acc(G, 5) | Acc(C) |
| InfoNCE | $0.0_{\pm0.02}$ | $0.0_{\pm0.00}$ | $98.9_{\pm0.83}$ | $0.0_{\pm0.01}$ | $0.0_{\pm0.01}$ | $99.1_{\pm0.55}$ | $0.1_{\pm0.15}$ | $0.0_{\pm0.00}$ | $98.5_{\pm0.56}$ | $0.36_{\pm0.08}$ | $99.97_{\pm0.03}$ |
| +LDS | $0.0_{\pm0.03}$ | $0.0_{\pm0.00}$ | $99.0_{\pm0.51}$ | $0.0_{\pm0.11}$ | $0.0_{\pm0.00}$ | $99.0_{\pm0.65}$ | $0.1_{\pm0.06}$ | $0.0_{\pm0.00}$ | $98.4_{\pm0.59}$ | $0.31_{\pm0.03}$ | $99.96_{\pm0.04}$ |
| +SLDS | $0.0_{\pm0.01}$ | $0.0_{\pm0.00}$ | $98.9_{\pm0.79}$ | $0.0_{\pm0.03}$ | $0.0_{\pm0.01}$ | $98.7_{\pm0.96}$ | $0.2_{\pm0.24}$ | $0.0_{\pm0.00}$ | $97.7_{\pm0.93}$ | $0.28_{\pm0.02}$ | $99.81_{\pm0.27}$ |
| EbC (lin.) | $70.9_{\pm2.60}$ | $54.1_{\pm4.63}$ | $20.2_{\pm1.94}$ | $70.8_{\pm2.62}$ | $54.0_{\pm4.66}$ | $19.8_{\pm2.14}$ | $59.7_{\pm8.10}$ | $39.8_{\pm13.66}$ | $22.3_{\pm4.63}$ | – | – |
| EbC | $99.7_{\pm0.22}$ | $99.7_{\pm0.25}$ | $99.1_{\pm0.72}$ | $99.8_{\pm0.05}$ | $99.7_{\pm0.04}$ | $99.2_{\pm0.69}$ | $99.8_{\pm0.03}$ | $99.7_{\pm0.06}$ | $98.5_{\pm0.69}$ | $99.91_{\pm0.05}$ | $74.04_{\pm1.91}$ |

## 5 Empirical Results

We apply EbC on a variety of group learning settings, summarized in Table 1. EbC is effective at recovering group structure both in a setting where we have precise access of the latent space (for verification), and scales to benchmarking datasets used in objective-centric learning. Across all baselines, only EbC is able to recover the structure with high fidelity: We reach an $R^2$ of $> 99\%$ in recovering the ground truth latent space on synthetic data, validating Thm 1a and substantially outperforming a linear baseline (60–70%). This result verifies that indeed $\mathbf{h}(\boldsymbol{x}) = \boldsymbol{Lx}$ up to a linear indeterminacy in practice. We verify the group structure through an $R^2$ between the projected sample and the ground-truth for synthetic data with perfect recovery of >98% (Thm 1b). As a verification, we compare to existing contrastive learning baselines which are well established at learning *invariant* representations, i.e., the content. Indeed, all considered baselines are effective at recovering the content information with accuracies typically >98%. On synthetic data, EbC matches this performance (>98%); on idSprites we encounter a trade-off and accuracy drops to 75%.

**EbC learns a representation of** $(R_m \times \mathbb{Z}_n \times \mathbb{Z}_n)$ **from image data.** We next evaluate EbC on idSprites. An optimal embedding space for the $R_m \times \mathbb{Z}_n \times \mathbb{Z}_n$ is a 3-torus, with the three individual groups represented along the three circular coordinates. Fig 4a depicts this (assumed) underlying structure of the latent space (which is not enforced in the real dataset or during training), along with the reconstructed embedding space of EbC. Note that the dimensions are related to each other; in Fig. 4b we show the dependency based on the y-, x- coordinate for a fixed angle. Qualitatively, this embedding space was stable also under variations of the content (shape and size of the object).

**EbC learns faithful representations of content and style under diverse conditions.** Next, we perform a fine-grained quantitative evaluation of the embedding space. In particular, we vary the properties of the indSprites dataset by varying the number of options for the content (shape and size) and the number of variations to learn from (translation and rotation) while keeping the overall dataset fixed. In Figure 4c we outline these options, reporting both the classification accuracy of the content, and the quality of the representation using our kNN metric. EbC recovers the group structure with high Top-5 accuracy close to 100% when performing an 1-over-20k kNN lookup for a range of number of contents. However, we notice that the content classification declines from around 80% to 20%. We report qualitative results of the KNN prediction in Appendix C.4.

**EbC is robust to over-parameterization** In Figure 4d, we vary the output dimension of the encoder (divided into 2 dimensions for the content, and 4–7 for the group). We observe that the group structure preserves a high predictive kNN above 99% and a stable content classification performance above 80%. However, this requires a sufficient number of samples for estimating the implicit representation: For $4\times$ the group dimensionality is required when single transitions are observed, and $8\times$ is required in the case of compound actions. Note that in practice, it is possible to even inform model hyperparameters on this metric, as it is available without any knowledge of the underlying ground truth structure of the data.

**EbC identifies group representations of** $O(n), GL(n),$ **and** $SO(n)$**.** Next, we investigate the applicability to more general group structures. For this, we generate additional synthetic data from more complex groups. $SO(n)$ is close to our example on idSprites, which is extended by $O(n)$ and $GL(n, R)$ as a fairly general and challenging example (Figure 5). Here we report results for $n = 3$,

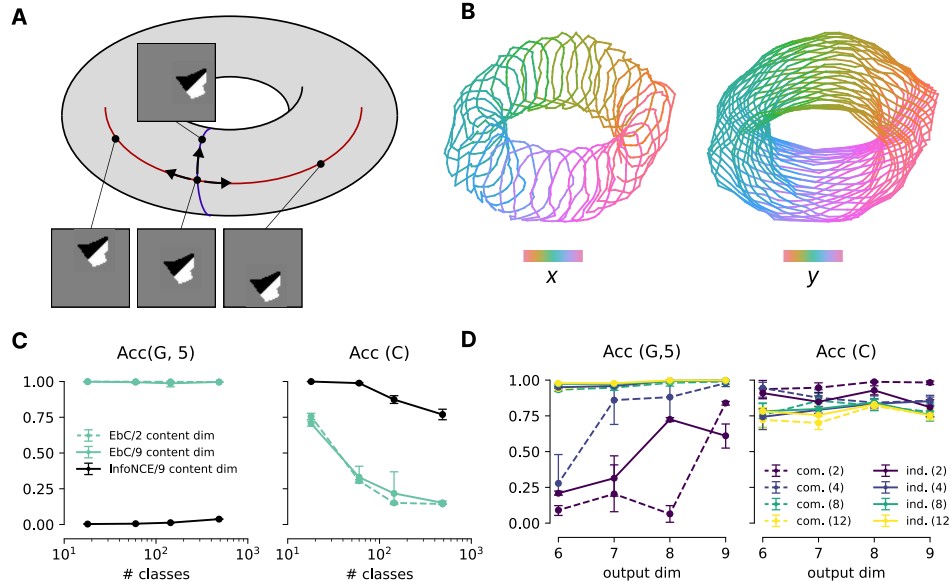

Figure 4: **EbC learns faithful representations of group actions**. A, a possible ground-truth latent space for idSprites. Depicted is $\mathbb{Z}_n \times \mathbb{Z}_n$. B, the actual embedding obtained by EbC, plotted for a fixed value of the orientation angle and colored in two views for variations in x- and y- direction. C, evaluation of representation quality and content accuracy across an increasing amount of classes. D, impact of observing the single groups (ind.) vs. a combination at each step in training (com.) across different output dimensions. Number in brackets is the number of samples in $Y$ per output dim.

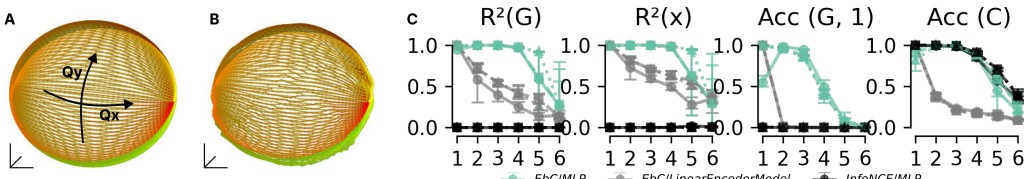

Figure 5: **Verification on simulated data**. A, traversals through the ground-truth latent space of our synthetic datasets. We manually constructed two orthogonal transformations $\boldsymbol{Q}_x, \boldsymbol{Q}_y$ perpendicular to each other to obtain a sphere. B, recovered latent space after non-linear demixing and optimal linear alignment. C, comparisons of EbC (green) against a linear baseline (gray) and an InfoNCE baseline (black) across an increasing number of mixing layers.

in Appendix C.1 we show additional results for higher dimensions. The complexity of the problem is additionally varied by generating increasingly non-linear datasets. This is expected to eventually degrade performance when a fixed-size dataset is used. We vary the number of mixing layers from n=2 to higher dimensions.

EbC is able to recover both a suitable vector space and an implicit group representation on these datasets with high fidelity. Figure 5a shows the ground truth latents, and Figure 5b shows a qualitative impression how the spherical latent space is recovered after unmixing. Quantatively, the R2(x) metric in Fig. 5c confirms the quality of this mapping up to n=4 mixing layers before we obtain degradation. Likewise, forward prediction through the group representation obtains high R2(G) up to 4 mixing layers, substantially outperforming the linear baseline. A relevant metric in practice is Acc(G) measuring forward prediction through a kNN based metric. In comparision to the InfoNCE baseline, we perform comparatively in identifying the class content across all group types (Acc(C)).

## 6 Further Analysis, Ablations, and Modeling Choices

Following experimental validation of our model and applicability to image data, we are interested in analyzing the empirical behavior of the loss function in more detail. In practice, an important

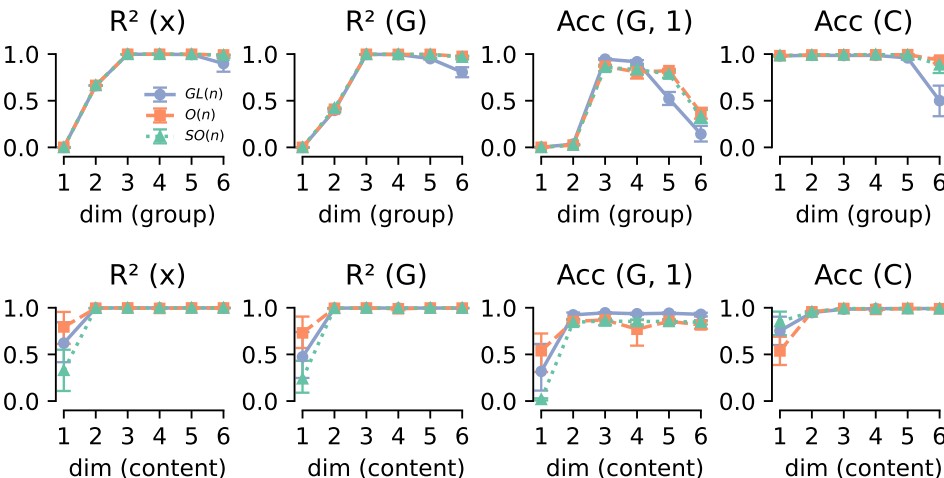

Figure 6: Robustness to model misspecification. In the ground-truth process, we allocate 3 dimensions to coordinates affected by the group action, and 3 dimensions to to content. Note, $R^2$ metrics reported here are not observable on real data, while Acc(G) is observable and Acc(C) is observable if a labeled test set is given.

hyperparameter is the parameterization of the embedding space, as well as the setup of the loss function. Further validation and ablation experiments can also be found in Appendix C.

**Identification of the correct latent dimensionality**  In practice, the true dimensionality of the latent vector space is not known, or even ambiguous. Algorithms for learning group structure should be ideally (1) robust to model misspecification and (2) provide a clear hyperparameter validation protocol for choosing the correct dimensionality. In Fig. 6, we report the properties of EbC under model misspecification. Both for the content and groups, we consider a 3d ground truth latent space, resulting in a total of six latent factors. In Figure 6, second row, we show performance variations as we specify the wrong content dimension. Except for a degenerative case for d=1, we obtain close to perfect recovery of both group structure and content.

In the first row of Figure 6 we misspecify the group dimensionality, which is more critical. We see a clear peak in 1-NN lookup performance (Acc(G)) as well as saturation for R2(x)/R2(G) prediction performance for d>3. Very crucially, the practically *observable* metric for hyperparameter selection, Acc(G), shows a clear peak at the correct dimensionality (d=3), making it a prime target for model selection (which we indeed did in the experiments showed so far, cf. the discussion in Appendix B). As before, with the correctly selected dimension, we obtain a close-to-perfect performance in both recovery of the latent space and the representation of the group.

**Analysis of optimal model parameters and hyperparameter selection**  With our fixed selection protocol, we finally consider model choices for the implementation of the algorithm outlined in Sec. 2. In particular, the choice of negatives for the set $S$ is relevant for satisfying the diversity conditions required for Theorem 1. Furthermore, we can extend the loss function to consider the property of an inverse element, i.e., that both the action $R(X, X')$ and $R(X', X)$ are valid representations. Note, we do not explicitly force the constraint that the respective matrices should be inverses of each other, making our approach still potentially applicable to structures not fulfilling all group axioms.

Table 2 depicts the different choices for both a symmetric loss and different negative distributions. On simpler groups, SO(n) and O(n), when our input vectors are normalized, the co-domain of the action will still be the sphere, making it less crucial to consider both $y$ and $y'$ as negative samples. In contrast, as we move to $GL(n)$, it is crucial to use both samples for training to avoid a performance drop. On O(n), GL(n), a symmetric loss additionally improves the performance.

# 7 Limitations

**Accuracy trade-off for many classes**  While EbC is able to convincingly estimate both the latent space and the group actions, we could construct a failure case in idSprites when class dimensions

Table 2: Model variations across different loss functions, negative samples, and groups.

| Sym. | $S$ | $SO_n$ | | $O_n$ | | $GL_n$ | |
| --- | --- | --- | --- | --- | --- | --- | --- |
| | | Acc(G, 1) | Acc(G, 5) | Acc(G, 1) | Acc(G, 5) | Acc(G, 1) | Acc(G, 5) |
| ✗ | both | $56.23_{\pm2.62}$ | $77.38_{\pm24.35}$ | $86.53_{\pm1.33}$ | $89.75_{\pm1.65}$ | $93.14_{\pm19.89}$ | $99.98_{\pm0.01}$ |
| ✗ | $y$ | $28.77_{\pm21.88}$ | $83.65_{\pm6.07}$ | $85.49_{\pm1.39}$ | $53.23_{\pm38.76}$ | $99.91_{\pm0.18}$ | $99.98_{\pm0.01}$ |
| ✗ | $y'$ | $73.44_{\pm4.40}$ | $79.52_{\pm18.17}$ | $87.08_{\pm1.49}$ | $95.91_{\pm1.49}$ | $96.83_{\pm9.15}$ | $99.98_{\pm0.01}$ |
| ✓ | both | $94.70_{\pm1.23}$ | $86.95_{\pm1.06}$ | $85.59_{\pm6.04}$ | $99.83_{\pm0.11}$ | $99.98_{\pm0.01}$ | $99.93_{\pm0.16}$ |
| ✓ | $y$ | $58.55_{\pm20.11}$ | $85.54_{\pm2.54}$ | $85.70_{\pm4.88}$ | $87.77_{\pm6.73}$ | $99.97_{\pm0.02}$ | $99.95_{\pm0.10}$ |
| ✓ | $y'$ | $94.44_{\pm1.06}$ | $84.31_{\pm6.99}$ | $83.61_{\pm7.88}$ | $99.84_{\pm0.09}$ | $99.89_{\pm0.25}$ | $99.85_{\pm0.35}$ |

increases to multiple hundreds or more. While we continue to estimate the group representation, the classification performance of the content decreases in these cases, even if the dimensionality of the latent space is matched to our baseline. A possible explanation is that EbC needs to implicitly learn a prototype of each class to estimate the coordinates affected by the group action, which might be harder to embed in the network than a classification objective alone.

**Style vs. content subspaces identifiability** Our proposed prior to separate content and style also has limitations. A clear separation can only be expected if the group dimensionality is chosen to be minimal. While we demonstrated that such a minimal dimensionality can be computed on real-world datasets, it comes with the additional computational burden of hyperparameter estimation, which might be prohibitive in practice. Future work might consider improved versions of our content/style subspace prior and provide theoretical guarantees on the optimal separation. Within the scope of this work, we showed a practical way to estimate the hyperparameters using the Acc(G) metric in the analysis section.

**Real-world data and baselines** Although we are the first to show that identifiable group representation learning from unlabeled observational data is feasible at all, our empirical analysis can be substantially extended. Firstly, on selected previous setups, more baseline comparisons could be conducted which is here limited due to the lack of extensive benchmarks. It will be particulary interesting to see how the algorithm scales to to full-scale image datasets like 3DIdent [56]. However, we do report additional experiments on real-world data in Appendix C.5.

**Scaling properties, empirical vs. theoretical** Our theoretical result mandates the use of $d + 1$ examples for estimation of the embedding space, and currently does not make a claim how convergence is influenced by limited data. In practice we observed that substantially more than $d + 1$ samples are required (about 6-8$\times$ gave good empirical results). In this light, our theoretical results can be extended to include bounds on the behavior with limited samples, although challenging and not every common in existing identifiability work. On the other hand, replacing our out-of-the-box least squares estimator by more stable and more adapted algorithms might further close the gap to the theoretical optimum. We report additional results on data efficiency in Appendix C.2.

## 8   Related Work

**Learning equivariant representations** Early work in equivariant representation learning focused on designing neural network architectures that are explicitly equivariant to a predefined class of transformations [3, 10, 27, 31, 42, 53]. These methods were powerful but require expert knowledge to design an inductive bias specific to particular groups. Inspired by the success of invariant self-supervised learning (I-SSL) methods [2, 13, 37, 54], a new paradigm emerged that sought to *learn* equivariant representations directly from data. The first wave of SSL-based methods assumed full knowledge of the group actions. Latent representations were learned either via encoder-only frameworks [4, 7, 12, 38] or via autoencoder-based frameworks [23, 24, 39]. Equivariance was encouraged either by predicting the parameters of the group action [4] or enforced by explicitly modeling the effects of group actions in latent space [7, 12, 23, 24, 38, 39]. The most recent advances tackle the more challenging problem of learning equivariance without explicit knowledge of the group actions. Both encoder-only approaches [15, 51, 52] and autoencoder-based approaches [28, 35, 36] have been proposed. The central innovation of these methods is that the representation of group actions $\boldsymbol{R}(g)$ is learned directly from data pairs $(\boldsymbol{y}, g \cdot \boldsymbol{y})$. For example, CARE [15] is an encoder-only, contrastive learning based approach with implicit $\boldsymbol{R}(g) \in O(n)$. STL [52] also uses an encoder-only

approach but enforces non-linear equivariant relationships $\phi(g \cdot \boldsymbol{y}) = \mu(\phi(\boldsymbol{y}), \boldsymbol{R}(g))$. NFT [28] adopts an auto-encoder framework with explicit linear group representations $\boldsymbol{R}(g) \in GL(n)$. While these methods mark important progress, they each leave open critical challenges: some are restricted to special classes of group representations, some require generative modeling of the input space, and none provide formal identifiability guarantees of the applied algorithm. While Koyama et al. [28] are the first to show that the problem itself is identifiable (i.e., there exists a non-trivial G-equivariant map $\phi : Y \mapsto X$ unique up to G-isomorphisms of the embedding space), they do not provide identifiability guarantees for the learning method itself.

**Identifiable nonlinear ICA** In nonlinear ICA, observed variables $\boldsymbol{y}$ are assumed to be generated from latent variables $\boldsymbol{x}$ via an unknown nonlinear "mixing" function $\mathbf{f}$. The goal is to recover $\mathbf{f}^{-1}$, or equivalently, estimate the latent variables $\boldsymbol{x}$ from the observed variables $\boldsymbol{y}$. Hyvärinen and Pajunen [20] showed that in contrast to linear ICA, the nonlinear ICA problem is, in general, *un*identifiable. Subsequent work has established that identifiability can be recovered by making additional assumptions about the data-generating process [18, 19, 21]. More recently, this theory has been connected to contrastive learning [40, 56], demonstrating that: (a) the choice of the conditional distribution of positive samples encodes the assumptions about the data-generating process, and (b) optimizing the InfoNCE loss [16, 37] recovers a solution to the nonlinear ICA problem. The identified representation is unique up to an indeterminacy that is directly determined by the assumed conditional distribution. Building on these advances, Laiz et al. [30] proposed DCL, which identifies latent representations under the assumption of linear or switching linear dynamical systems. Since G-equivariant representations also rely on a linear relationship between $(\boldsymbol{x}, \boldsymbol{x}')$ or $(\boldsymbol{x}_t, \boldsymbol{x}_{t+1})$) in sequential data, there exists a natural connection between DCL and the equivariant representation learning methods discussed above. Our work leverages this connection, combining the identifiability guarantees of nonlinear ICA with the goals of equivariant representation learning.

# 9 Conclusion

We proposed the first identifiable learning algorithm for group representations from unlabeled data without relying on generative models. We demonstrated, theoretically and empirically, that latent spaces of datasets with underlying group structure can be estimated from observational data. Crucially, the assumptions of our algorithms match datasets encountered in engineering and science. The supervisory signal merely requires knowledge that "the same" action is applied to a collection of samples. We confirmed that under these settings and mild overall assumptions on sufficient variety in the dataset, few of such pairs are sufficient to estimate the underlying ground truth latent space in which the group actions are faithfully represented as a linear transformation. We see wide applicability of this approach especially in computer vision (e.g., for figure-ground segmentation), robotics (for learning affordances and planning), and biology and biomedicine (for modeling the effects of longitudinal data and perturbation effects), among other fields.

## Acknowledgments and Disclosure of Funding

We thank Susanne Keller for in-depth discussions about the model setup and related work. We thank Luisa Eck for discussions on the theory. We thank the anonymous reviewers and the area chair at NeurIPS 2025 for their constructive feedback to improve our manuscript. This work was supported by a Google PhD fellowship to StS. This work was supported by the German Research Foundation (DFG): SFB 1233, Robust Vision: Inference Principles and Neural Mechanisms, project number: 276693517 and by Open Philanthropy Foundation funded by the Good Ventures Foundation. MB is a member of the Machine Learning Cluster of Excellence, funded by the Deutsche Forschungsgemeinschaft (DFG, German Research Foundation) under Germany's Excellence Strategy – EXC number 2064/1 – Project number 390727645. This work was supported by the Helmholtz Association's Initiative and Networking Fund on the HAICORE@KIT and HAICORE@FZJ partitions.

## Author contributions

**TS**: Methodology, Software, Investigation, Writing–Editing; **StS**: Conceptualization, Methodology, Formal analysis, Writing–Original Draft, **MB**: Conceptualization, Writing–Editing.

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

# Supplementary Material

# A  Proof of Theorem 1

In this chapter, we provide a proof for Theorem 1 in the main paper. We assume the following data generating process:

**Definition 1** (Data generating process). Let $\boldsymbol{x} \in V \subseteq \mathbb{R}^d$ be a vector describing the ground truth latent components, let $g$ be an element of a group $G$, and let $\mathbf{f} : V \to \mathbb{R}^D$ be an injective map. Assume that for each group element $g$, we have at least $M$ pairs of samples $(\boldsymbol{y}_j, \boldsymbol{y}'_j)$ with

$$\boldsymbol{y}_i = \mathbf{f}(\boldsymbol{x}_i), \qquad \boldsymbol{y}'_i = \mathbf{f}(\boldsymbol{R}(g)\boldsymbol{x}_i) \qquad i = 1, \ldots, M \tag{15}$$

where $\boldsymbol{R} : G \to \mathrm{GL}(d, \mathbb{R})$ is a representation of $G$ on $V$.

If we think of $\boldsymbol{x}$ containing an invariant part (the content), the theory by Von Kügelgen et al. [47] already covers the case for identifying the content component of $\boldsymbol{x}$ when considering the group actions as augmentations. We will therefore not explicitly discuss the discovery of the invariant part of the embedding, but focus our following statements on learning equivariant representations.

We extend the canonical discriminative form introduced by Roeder et al. [40] to arrive at the form

$$p_{\boldsymbol{u},\boldsymbol{v},\alpha,\beta}(\boldsymbol{y} \mid \boldsymbol{x}, S) = \frac{\exp\left(\boldsymbol{u}(\boldsymbol{x})^\top \boldsymbol{v}(\boldsymbol{y}) + \alpha(\boldsymbol{x}) + \beta(\boldsymbol{y})\right)}{\sum_{\boldsymbol{y}' \in S} \exp\left(\boldsymbol{u}(\boldsymbol{x})^\top \boldsymbol{v}(\boldsymbol{y}') + \alpha(\boldsymbol{x}) + \beta(\boldsymbol{y}')\right)}. \tag{16}$$

The functions $\boldsymbol{u}, \boldsymbol{v} : \mathbb{R}^d \to \mathbb{R}^d$, $\alpha, \beta : \mathbb{R}^d \to \mathbb{R}$ can have shared underlying structure (e.g., are parameterized by the same neural network) and are not necessarily independent. We extend the diversity conditions [40]:

**Definition 2** (Diversity Conditions, adapted from Roeder et al. [40]). Assume a dataset of tuples $(\boldsymbol{x}, \boldsymbol{y}, S)$ and let $p_{\mathrm{data}}(\boldsymbol{x}, \boldsymbol{y}, S)$ be a distribution over this (possibly discrete) dataset. We then require the following properties:

1. For any $\boldsymbol{x}, S$ we can find at least $N$ pairs of points $(\boldsymbol{y}_A, \boldsymbol{y}_B)$ with $p_{\mathrm{data}}(\boldsymbol{x}, \boldsymbol{y}_A, S) > 0$ and $p_{\mathrm{data}}(\boldsymbol{x}, \boldsymbol{y}_B, S) > 0$ such that the finite differences $\boldsymbol{v}(\boldsymbol{y}_A^i) - \boldsymbol{v}(\boldsymbol{y}_B^i)$ are linearly independent.

2. For any $\boldsymbol{y}$, we can find at least $N$ pairs of points $\boldsymbol{x}_A, \boldsymbol{x}_B$ and their sets of negative samples $S_A, S_B$ such that $\boldsymbol{y} \in S_A \cap S_B$ is in the negative sets for both examples, and the finite differences $\boldsymbol{u}(\boldsymbol{x}_A^i) - \boldsymbol{u}(\boldsymbol{x}_B^i)$ are linearly independent.

Both assumptions need to be assured through design of the sampling procedure and initialization of the networks. In particular, the first condition requires that sufficient variation is present among negative examples. Similar to Roeder et al. [40], since $\boldsymbol{v}$ is a randomly initialized neural network, it is expected that the variability is met as long as variability in the ground truth factors is given. The second condition requires that negative samples sufficiently often co-occur with enough variation in the reference examples. In addition, this condition requires $\hat{\boldsymbol{R}}$ to be full-rank, which we ensure by adding at least $d$ samples for estimating a $d \times d$ matrix.

We can now re-state the theorem and discuss the proof strategy. Please note that we relate the embedding space according to the commutative diagram in Figure 1: We consider the relations between samples in the original vector space $V$ vs. the recovered factor space, which is the co-domain of first applying the mixing and then the de-mixing functions, $\mathbf{h} := \phi \circ \mathbf{f}$, $\mathbf{h} : V \to V$. We can state:

**Theorem 1** (Identifiability of the Group Representation). *Assume the following:*

*(1) $\mathbf{f}$ is an injective mixing function, $g \in G$ is a group element according to Def. 3.*
*(2) The model $\phi : \mathbb{R}^D \to V$ satisfies the diversity conditions in Def. 4.*
*(3) $\boldsymbol{R}$ is a representation of $G$, and $\hat{\boldsymbol{R}}$ is the implicit representation of the group (Eq. 2) such that $\hat{\boldsymbol{R}}(\boldsymbol{X}, g\boldsymbol{X}) = \boldsymbol{R}(g)$ for pairs of transformed samples $(\boldsymbol{X}, g\boldsymbol{X})$ and group actions $g \in G$.*

*Define $\mathbf{h} := \phi \circ \mathbf{f}$. Then, for matching conditional distributions $p_\phi = p_{\mathbf{f}^{-1}}$, for all points $\boldsymbol{x}$ and group actions $g$ in the dataset:*

*(a) We recover the original vector space $\mathbf{h}(\boldsymbol{x}) = \boldsymbol{L}\boldsymbol{x}$, up to an ambiguity $\boldsymbol{L} \in GL(d)$.*
*(b) We recover a representation of the group, $\hat{\boldsymbol{R}}(\mathbf{h}(\boldsymbol{X}), \mathbf{h}(g\boldsymbol{X})) = \boldsymbol{L}\boldsymbol{R}(g)\boldsymbol{L}^{-1}$.*

As outlined in the main paper, the proof proceeds as follows: First, we extend the canonical discriminative form of Roeder et al. [40] towards a more general setting, giving affine instead of linear identifiability (§A.1). Then, we show that our implicit group representation admits (§A.2). Finally, we leverage these results to arrive at the indeterminacies reported in the theorem (§A.3).

## A.1 Extended canonical discriminative form

The following proposition is only a slight modification from Eq. 1 in Roeder et al. [40]. The key difference is the introduction of the scalar potential functions $\alpha$ and $\beta$, making the canonical discriminative form more flexible. In particular, it now admits a mean squared error loss function.

Under these assumptions, we can state:

**Theorem 2** (Generalized Canonical Discriminative Form). *Let $\boldsymbol{u}, \boldsymbol{v} : \mathbb{R}^d \to \mathbb{R}^d$, $\alpha, \beta : \mathbb{R}^d \to \mathbb{R}$ be functions satisfying Def. 2. In particular, $\boldsymbol{u}, \boldsymbol{v}, \alpha, \beta$ can have shared underlying structure and are not necessarily independent. Then, the probabilistic model of the form*

$$p_{\boldsymbol{u},\boldsymbol{v},\alpha,\beta}(\boldsymbol{y} \mid \boldsymbol{x}, S) = \frac{\exp\left(\boldsymbol{u}(\boldsymbol{x})^\top \boldsymbol{v}(\boldsymbol{y}) + \alpha(\boldsymbol{x}) + \beta(\boldsymbol{y})\right)}{\sum_{\boldsymbol{y}' \in S} \exp\left(\boldsymbol{u}(\boldsymbol{x})^\top \boldsymbol{v}(\boldsymbol{y}') + \alpha(\boldsymbol{x}) + \beta(\boldsymbol{y}')\right)} \tag{17}$$

*with $\boldsymbol{y} \in S$ is identifiable up to an affine indeterminacy, i.e., for two models $(\boldsymbol{u}', \boldsymbol{v}', \alpha', \beta')$ and $(\boldsymbol{u}^*, \boldsymbol{v}^*, \alpha^*, \beta^*)$ and their respective distributions $(p', p^*)$, we have for all $\boldsymbol{x}, \boldsymbol{y}$ with $p_{\mathrm{data}}(\boldsymbol{x}, \boldsymbol{y}, S) > 0$,*

$$p' = p^* \implies \begin{cases} \boldsymbol{u}'(\boldsymbol{x}) & = \boldsymbol{A}\boldsymbol{u}^*(\boldsymbol{x}) + \boldsymbol{c} \\ \boldsymbol{v}'(\boldsymbol{y}) & = \boldsymbol{B}\boldsymbol{v}^*(\boldsymbol{y}) + \boldsymbol{d} \end{cases} \tag{18}$$

*for two invertible $d \times d$ matrices $\boldsymbol{A}$ and $\boldsymbol{B}$ and vectors $\boldsymbol{c}, \boldsymbol{d} \in \mathbb{R}^d$.*

*Proof.* The proof technique adopts the strategy from Roeder et al. [40] for the indeterminacy of $\boldsymbol{u}$, and re-applies this to the indeterminacy of $\boldsymbol{v}$. The approach considers finite differences on the log probabilities for $p, p^*$ with

$$\log p(\boldsymbol{y}|\boldsymbol{x}, S) = \boldsymbol{u}(\boldsymbol{x})^\top \boldsymbol{v}(\boldsymbol{y}) + \alpha(\boldsymbol{x}) + \beta(\boldsymbol{y}) - Z(\boldsymbol{x}, S) \tag{19}$$

$$= \boldsymbol{u}(\boldsymbol{x})^\top \boldsymbol{v}(\boldsymbol{y}) + \tilde{\alpha}(\boldsymbol{x}, S) + \beta(\boldsymbol{y}). \tag{20}$$

**Part 1: Affine identifiability of $\boldsymbol{u}$.** For any $\boldsymbol{x}$ in the data distribution, and corresponding $S$, select two points $\boldsymbol{y}_A, \boldsymbol{y}_B \in S$. For the points, it holds that

$$\log p(\boldsymbol{y}_A|\boldsymbol{x}, S) - \log p(\boldsymbol{y}_B|\boldsymbol{x}, S) = \log p^*(\boldsymbol{y}_A|\boldsymbol{x}, S) - \log p^*(\boldsymbol{y}_B|\boldsymbol{x}, S) \tag{21}$$

Let $\Delta \boldsymbol{v}_{AB} := \boldsymbol{v}(\boldsymbol{y}_A) - \boldsymbol{v}(\boldsymbol{y}_B)$, $\Delta \beta_{AB} := \tilde{\beta}(\boldsymbol{y}_A) - \beta(\boldsymbol{y}_B)$ to obtain

$$\Delta \boldsymbol{v}_{AB}^\top \boldsymbol{u}(\boldsymbol{x}) + \Delta \beta_{AB} = (\Delta \boldsymbol{v}_{AB}^*)^\top \boldsymbol{u}^*(\boldsymbol{x}) + \Delta \beta_{AB}^* \tag{22}$$

By assumption (1), we can find $N$ such pairs with linearly independent $\Delta \boldsymbol{v}_{AB}, \Delta \boldsymbol{v}_{AB}^*$, which lets us re-write the equation using the full-rank matrices $\boldsymbol{L}, \boldsymbol{L}^*$

$$\boldsymbol{L}^\top \boldsymbol{u}(\boldsymbol{x}) + \boldsymbol{c} = (\boldsymbol{L}^*)^\top \boldsymbol{u}^*(\boldsymbol{x}) + \boldsymbol{c}^* \tag{23}$$

$$\boldsymbol{u}(\boldsymbol{x}) = (\boldsymbol{L}^* \boldsymbol{L}^{-1})^\top \boldsymbol{u}^*(\boldsymbol{x}) + (\boldsymbol{c}^* - \boldsymbol{c}) \tag{24}$$

which requires the map from $\boldsymbol{u}$ to $\boldsymbol{u}^*$ to be affine, concluding the first part of the proof.

**Part 2: Affine identifiability of $\boldsymbol{v}$.** For a given $\boldsymbol{y}$, find $\boldsymbol{x}_A, \boldsymbol{x}_B, S_A, S_B$ such that $\boldsymbol{y} \in S_A \cap S_B$, and note that $S_A = S_B$ is possible but not required. Then we consider the finite differences:

$$\log p(\boldsymbol{y}|\boldsymbol{x}_A, S_A) - \log p(\boldsymbol{y}|\boldsymbol{x}_B, S_B) = \log p^*(\boldsymbol{y}|\boldsymbol{x}_A, S_A) - \log p^*(\boldsymbol{y}|\boldsymbol{x}_B, S_B) \tag{25}$$

Let $\Delta \boldsymbol{u}_{AB} := \boldsymbol{u}(\boldsymbol{x}_A) - \boldsymbol{u}(\boldsymbol{x}_B)$, $\Delta \tilde{\alpha}_{AB} := \tilde{\alpha}(\boldsymbol{x}_A, S) - \tilde{\alpha}(\boldsymbol{x}_B, S)$ to obtain

$$\Delta \boldsymbol{u}_{AB}^\top \boldsymbol{v}(\boldsymbol{y}) + \Delta \tilde{\alpha}_{AB} = (\Delta \boldsymbol{u}_{AB}^*)^\top \boldsymbol{v}^*(\boldsymbol{y}) + \Delta \tilde{\alpha}_{AB}^* \tag{26}$$

Stacking multiple conditions gives

$$\boldsymbol{L}^\top \boldsymbol{v}(\boldsymbol{y}) + \boldsymbol{c} = (\boldsymbol{L}^*)^\top \boldsymbol{v}^*(\boldsymbol{y}) + \boldsymbol{c}^* \tag{27}$$

$$\boldsymbol{v}(\boldsymbol{y}) = (\boldsymbol{L}^* \boldsymbol{L}^{-1})^\top \boldsymbol{v}^*(\boldsymbol{y}) + (\boldsymbol{c}^* - \boldsymbol{c}) \tag{28}$$

which requires the map from $\boldsymbol{v}$ to $\boldsymbol{v}^*$ to be affine, concluding the proof. $\square$

## A.2 Properties of the implicit group representations

We next consider properties of the implicit group representation defined in Sec. 2, given by

$$\hat{R}(X, X') = \min_{R \in \mathrm{GL}(d)} \|X' - XR^\top\|_F^2 \Leftrightarrow \hat{R}(X, X') = (X^\top X)^{-1}(X^\top X'). \qquad (29)$$

**Lemma 1** (Equivariance of implicit group representations). *Let $X, X' \in \mathbb{R}^{m \times n}$, $m \geq n$ have rank $n$. Define $\hat{R} : \mathbb{R}^{m \times n} \times \mathbb{R}^{m \times n} \to \mathbb{R}^{n \times n}$ as*

$$\hat{R}(X, X') = \arg\min_L \|X' - XL^\top\|_F^2 \qquad (30)$$

*and let $\mathbf{f}(X) = XA^\top + \mathbf{1}_m b^\top$ be an invertible affine transform applied to each row of $X$, then we have*

$$\hat{R}(\mathbf{f}(X), \mathbf{f}(X')) = A\hat{R}(X, X')A^{-1}. \qquad (31)$$

*Proof.* The solution to the least squares problem can be re-written using the normal equations,

$$\hat{R}(X, X') = \arg\min_L \|X' - XL^\top\|_F^2 \quad \Leftrightarrow \quad (X^\top X)\hat{R}(X, X')^\top = X^\top X' \qquad (32)$$

Let $\tilde{X} = XA^\top$ and $\tilde{X}' = X'A^\top$. Then, inserting the definition of $\mathbf{f}$ gives

$$\hat{R}(\mathbf{f}(X), \mathbf{f}(X')) = \arg\min_L \|\tilde{X}' + \mathbf{1}_m b^\top - \tilde{X}L^\top - \mathbf{1}_m b^\top\|_F^2 \qquad (33)$$

$$= \arg\min_L \|\tilde{X}' - \tilde{X}L^\top\|_F^2 \qquad (34)$$

which is equivalent to solving the normal equations

$$\left(\tilde{X}^\top \tilde{X}\right) \hat{R}(\mathbf{f}(X), \mathbf{f}(X'))^\top = \tilde{X}^\top \tilde{X}' \qquad (35)$$

Substituting back in terms of $X$ and $A$:

$$AX^\top XA^\top \hat{R}(\mathbf{f}(X), \mathbf{f}(X'))^\top = AX^\top X'A^\top \qquad (36)$$

$A$ is invertible by assumption and it follows

$$X^\top X \left[A^\top \hat{R}(\mathbf{f}(X), \mathbf{f}(X'))^\top A^{-\top}\right] = X^\top X' \qquad (37)$$

Note that $X^\top X \in \mathbb{R}^{n \times n}$ has full rank by the assumption on $X$. Then, we get

$$A^\top \hat{R}(\mathbf{f}(X), \mathbf{f}(X'))^\top A^{-\top} = (X^\top X)^{-1} X^\top X' = \hat{R}(X, X')^\top \qquad (38)$$

Taking the transpose and re-arranging yields

$$\hat{R}(\mathbf{f}(X), \mathbf{f}(X')) = A\hat{R}(X, X')A^{-1} \qquad (39)$$

concluding the proof. $\square$

## A.3 Proof of Theorem 1

*Proof.* We have $p_\phi = p_{\mathbf{f}^{-1}}$. After rewriting Eq. 3 in terms of $x, X, X'$ instead of $y = \mathbf{f}(x), Y = \mathbf{f}(X), Y' = \mathbf{f}(X')$, our model follows the generalized canonical discriminative form (Proposition 2) with the following parametrization for the data likelihood $p_{\mathbf{f}^{-1}}$:

$$u^*(x, X, X') = \hat{R}(X, X')x = R(g)x \qquad (40)$$

$$v^*(gx) = gx. \qquad (41)$$

To specify the model likelihood $p_\phi$, we introduce the shorthand $\mathbf{h} := \phi \circ \mathbf{f}$ which maps samples from the ground truth latent space to the recovered latent space (e.g., the composition of the data generating process and feature encoder) and obtain

$$u'(x, X, X') = \hat{R}(\mathbf{h}(X), \mathbf{h}(X'))\mathbf{h}(x) \qquad (42)$$

$$v'(x') = \mathbf{h}(x') \qquad (43)$$

By assumption $p_\phi = p_{\mathbf{f}^{-1}}$. From Proposition 2 it then follows that

$$u'(x, X, X') = A u^*(x, X, X') + c \tag{44}$$

$$v'(x') = B v^*(x') + d. \tag{45}$$

where $A, B \in \mathrm{GL}(d)$ and $c, d \in \mathbb{R}^d$. Inserting model and data generating process yields

$$\mathbf{h}(x') = B x' + d \tag{46}$$

$$\hat{R}(\mathbf{h}(X), \mathbf{h}(X')) \mathbf{h}(x) = A \hat{R}(X, X') x + c \tag{47}$$

Since $\mathbf{h}$ is affine, we insert the first into the second equation and invoke Lemma 1 to arrive at

$$B \hat{R}(X, X') B^{-1} (B x + d) = A \hat{R}(X, X') x + c \tag{48}$$

and re-arranging yields

$$B \hat{R}(X, X') x + B \hat{R}(X, X') d = A \hat{R}(X, X') x + c \tag{49}$$

$$(B - A) \hat{R}(X, X') x = c - B \hat{R}(X, X') d \tag{50}$$

This is an equation of the form $Ux = v$. Assuming that we observe $x$ such that matrix collecting all $x$ has full rank, i.e. all latent dimensions vary, it follows that $U = 0$, $v = 0$, hence

$$(B - A) \hat{R}(X, X') = 0 \tag{51}$$

$$B \hat{R}(X, X') d = c \tag{52}$$

and inserting $\hat{R}(X, X') = R(g)$ gives

$$(B - A) R(g) = 0 \tag{53}$$

$$c = B R(g) d \tag{54}$$

Since $R(g)$ has full rank, from the first equation it follows that $A = B$. For the second equation, since the left hand side is independent of $g$ and all matrices have full rank, we can only admit the trivial solution where $c = d = 0$. It follows that

$$\mathbf{h}(x) = B x \quad \forall x \in S \tag{55}$$

$$\hat{R}(\mathbf{h}(X), \mathbf{h}(X')) = B \hat{R}(X, X') B^{-1} \tag{56}$$

for an invertible matrix $B$, which concludes the proof. $\qquad\square$

# B  Additional Experimental Details

In this section, we provide further details on the algorithm and other experimental details. As an extension of this section, we would like to point to our figure and results repository (https://github.com/dynamical-inference/NeurIPS2025_Schmidt) as well as to our code repository (https://github.com/dynamical-inference/ebc) as the reference for all experimental details. The figure repository contains the code to plot the figures, but more importantly, it also contains all the raw results and all training and data generation parameters of all model runs that went into the respective figures. Each of those parameters can easily be traced back to their definition in the main code base repository.

## B.1  Experimental protocol

Here we describe the default dataset generation and training protocol of our experiments first, and then provide additional details for any deviation from these default parameters that may exist in any of the figures.

### B.1.1  Dataset Generation

**Synthetic Group Data**  We consider three types of synthetic datasets following the data generating process of Theorem 1. Group elements are taken from a subgroup of the special orthogonal group $SO(n)$, the orthogonal group $O(n)$ and the general linear group $GL(n)$:

1. We sample a random group elements $\boldsymbol{R}(g) \sim p_G(g)$ from $G \in \{SO(n), O(n), GL(n)\}$.
2. We sample $m$ (with $m > n$) random vectors $\boldsymbol{x} \in \mathbb{S}^n$ from the n-dimensional unit sphere.
3. We sample the content component $\boldsymbol{c} \in C$ from a finite set of vectors $C \subset S^d$.
4. From $(\boldsymbol{R}(g), \{\boldsymbol{x}_i\}^m, \boldsymbol{c})$ we generate $\boldsymbol{Y} = \{\boldsymbol{y}_i, \boldsymbol{y}'_i\}^m$,
   where $\boldsymbol{y}_i = \mathbf{f}(\boldsymbol{x}_i, \boldsymbol{c})$ and $\boldsymbol{y}'_i = \mathbf{f}(\boldsymbol{R}(g)\boldsymbol{x}_i, \boldsymbol{c})$.
5. We repeat this sampling procedure until we reach the desired dataset size.

To sample from the $O(n)$ and $SO(n)$ group, we sample from the Haar distribution [34], making use of the SciPy [46] package. To sample from $GL(n)$ we implement a rejection sampling procedure. We first generate a random $\boldsymbol{A} \in \mathbb{R}^{n \times n}$ matrix with values $a_{i,j}$ from a standard normal distribution. Then we compute the determinant $det(\boldsymbol{A})$ & the "identity error" $I_{err} = |\boldsymbol{I} - AA^{-1}|^F$ and we reject A if $|det(A)| < 0.2$ or $det(A) > 10$ or $I_{err} > 10^{-3}$, otherwise we accept. The nonlinear injective mixing function $\mathbf{f}$ is parameterized by a random 3-layer MLP and sampled via rejection sampling to fulfill the injectivity assumption as is done in [18, 30]. The MLP is followed by a random linear map to a 50-dimensional space. For the full dataset, we sample a maximum of 1000 matrices $\boldsymbol{R}(g)$ and a total of 1M pairs $(\boldsymbol{y}, \boldsymbol{y}')$. Unless stated otherwise, we choose $n = 3$, $d = 3$, $|C| = 100$.

**Infinite dSprites**  We leverage infinite dSprites [8, idSprites], an extension of dSprites [33] for validation on more diverse visual data. infinite dSprites is an extension of the original dSprites dataset, which allows to generate variations of dSprites dataset. We subsample all images to a resolution of $64 \times 64$. The dataset allows rich variations of the content and ensures that the resulting objects do not have any symmetries. By default we use the same number of varying factors as in dSprites, namely 3 random shapes of 6 different sizes. For our purposes we consider the combination of shape and size the content. Each of these objects may then be oriented in 40 different ways or have one of $32 \times 32$ x or y position translations. We consider the orientations and translations to be our groups of interest and sample the data such that we get closed groups. By default, we sample the dataset such that any paired sample $(\boldsymbol{y}, \boldsymbol{y}\prime)$ is undergoing a combination of these three types of transformations.

### B.1.2  Dataset Split

Before training on the synthetic group dataset, we perform a hold-out split in terms of the group actions $g_i$ into a training, validation, and test dataset such that we get a 80/10/10 split of the group actions. For idSprites, we randomly sample approx. 20k group actions during training and sample another approx. 90k group actions for validation and test respectively. When computing metrics that require to fit another model, such as a KNN, Linear Regression or Logistic Regression model, we split the validation or test dataset again 50/50 and report the metric on the hold-out set.

### B.1.3 Model Training

**Estimating $\boldsymbol{R}(g)$** To estimate $\hat{\boldsymbol{R}}(\boldsymbol{X}, \boldsymbol{X}')$ we use $k \cdot n$ additional samples where $n$ is the assumed group dimensionality of the embedding space and $k$ is the samples per action factor. We fit $\hat{\boldsymbol{R}}(\boldsymbol{X}, \boldsymbol{X}')$ via linear least squares (PyTorch's "gels" solver). By default, we do not compute gradients through the linear least squares estimation to save on compute time. By default for idSprites we use $k = 12$ and for the synthetic group dataset we use $k = 4$.

**Encoder Model** For the feature encoder $\phi$ we use an MLP with three layers and hidden unit size of 512 for idSprites and 128 for the synthetic group dataset. The output dimension of the MLP is $n + d$ where $n$ is the assumed group dimensionality of the embedding space and $d$ the assumed content dimensionality. Before passing the idSprites images to the MLP we scale them to $[-1, 1]$ and flatten the image. By default for idSprites we assume group dimensionality $n = 7$ and content dimensionality $d = 2$. For the synthetic group dataset we use the same dimensionalities $n$ and $d$ as are used for the data generation.

**Loss Function & Optimizer** As loss we minimize the negative mean likelihood (Eq. 3) and for each sample compute the loss in both directions $L(\boldsymbol{y}, \boldsymbol{y}')$ and $L(\boldsymbol{y}', \boldsymbol{y})$. As negatives $S$, we sample uniformly from all available samples in the dataset. We use 1024 positive pairs and 16k ($2^{12}$) negative samples in each batch and train each model for 20k iterations. We train the encoder via gradient descent using the Adam optimizer [25] with a learning rate of $1 \times 10^{-3}$.

**Error bounds** For our main results, to compute error bounds or standard deviation across our metrics, we generate each dataset with three different seeds and fit each model three times with different seeds during training. For variations and ablation experiments extending our main results, we only fit a single model across the three different dataset seeds. We refer to the Figure & Results repository for details on the exact number of seeds for every experiment.

## B.2 Hyperparameter Selection

We use the $Acc(G, 1)$ metric for any and all hyperparameter selection, since this is an observable metric and a good proxy for the identifiability metrics $R^2(G)$ (see Figure 8) and, by extension, for $R^2(x)$. Additionally, when we perform explicit hyperparameter selection and do not present results from experimental variations or ablation studies, we display the metrics computed on the test set instead of the metrics from the validation set, which were used for hyperparameter selection. Otherwise, we show the metrics on the validation set. Again, we refer to the Figure and Results repository for more details.

## B.3 Third-Party License Information

We used the infinite dSprites (idSprites; 8) dataset available at `https://github.com/sbdzdz/idsprites` in the pip-installable version v1.0.1 (MIT License).

## B.4 Compute Resources

Experiments were carried out on a compute cluster with A100 cards with 40Gb VRAM. On each card, we ran 2–6 experiments simultaneously, depending on the dataset size. The run time for individual experiments trained for 20k steps varied between approximately 10 minutes and one hour depending on the experiment configuration, the most important factors that impact the train time being batch sizes, specifically the number of negative samples, and the $k \cdot n$ number of samples used to fit the linear least squares estimator.

## C   Additional Experimental Results

### C.1   Recovery of group structure for $n > 3$.

We extend our results from the paper towards higher dimensions, and consider $SO(n), O(n), GL(n)$ for $n = 3, 5, 7, 9$. Results are depicted in Fig. 7. We mirror the main paper results with close to perfect reconstruction scores both for the latent space and the group representation for $SO(n), O(n)$, but notice a drop in performance for $GL(n)$ as dimensionality increases beyond 5. In this case, we can recover the performance for $n = 7$ if we allow to backpropagate through the least-squares solver. As the dimension grows to $n = 9$, the model is no longer able to discover the full group structure, likely due to a mismatch between the complexity of $GL(9)$ and the size of the dataset/variation present.

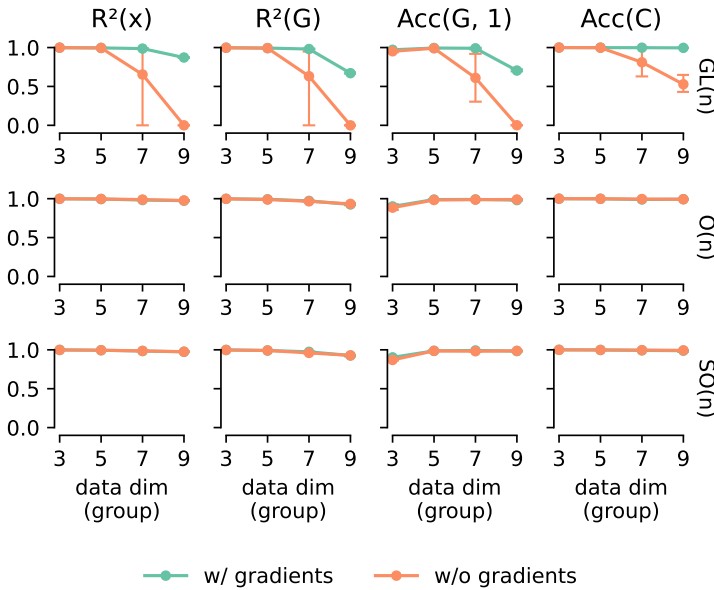

Figure 7: **Higher Latent Dimensions for Synthetic Group Data**. Comparison of EbC across an increasing number of true latent dimensions of the synthetic group dataset. We show results for all group types $GL(n)$, $O(n)$, $SO(n)$ with $n$ being on the x axis and fixed content dimension $d = 5$. Additionally, we indicate whether we compute gradients through the linear least squares estimator or not.

### C.2   Data efficiency

We analyze the data efficiency of our method by varying three key hyperparameters while keeping all other settings identical to those for $n = 3$ in Table 1 of the main paper. We test across $SO(3), O(3)$, and $GL(3)$ by:

1. Varying the total dataset size.
2. Varying the number of negative samples used in the contrastive loss.
3. Varying the number of samples per action ($k$) used for the linear least-squares fit of $\hat{\boldsymbol{R}}(g)$.

**Discussion (Dataset Size):** The results in Table 3 show that our method is highly robust to smaller dataset sizes. Even with 20x less data (50k vs 1000k), the identifiability of the equivariant representation ($R^2(x)$ and $R^2(G)$) remains nearly perfect, staying above 99.6% for all groups. The primary effect of smaller datasets is a moderate drop in accuracy for the invariant part of the embedding ($Acc(C)$).

Table 3: Data efficiency: Reduced dataset size. We evaluate EbC trained on datasets ranging from 50k to 1M samples (1000k). The 1000k setting replicates the main paper's results. We report mean and standard deviation over 5 runs for $n = 3$.

| Group Type | Dataset Size / Metric | 1000k | 100k | 500k | 50k |
|---|---|---|---|---|---|
| SO(n) | R²(x) | $99.81_{\pm0.04}$ | $99.79_{\pm0.06}$ | $99.75_{\pm0.22}$ | $99.80_{\pm0.03}$ |
| | R²(G) | $99.75_{\pm0.04}$ | $99.71_{\pm0.07}$ | $99.66_{\pm0.28}$ | $99.72_{\pm0.05}$ |
| | Acc(C) | $99.19_{\pm0.69}$ | $97.30_{\pm2.11}$ | $98.97_{\pm0.72}$ | $95.25_{\pm1.96}$ |
| O(n) | R²(x) | $99.80_{\pm0.06}$ | $99.81_{\pm0.03}$ | $99.79_{\pm0.12}$ | $99.73_{\pm0.16}$ |
| | R²(G) | $99.73_{\pm0.06}$ | $99.74_{\pm0.03}$ | $99.72_{\pm0.14}$ | $99.65_{\pm0.19}$ |
| | Acc(C) | $99.06_{\pm0.87}$ | $97.90_{\pm1.46}$ | $98.92_{\pm0.76}$ | $94.57_{\pm1.68}$ |
| GL(n) | R²(x) | $99.83_{\pm0.02}$ | $99.82_{\pm0.03}$ | $99.82_{\pm0.02}$ | $99.82_{\pm0.02}$ |
| | R²(G) | $99.72_{\pm0.05}$ | $99.71_{\pm0.05}$ | $99.71_{\pm0.04}$ | $99.68_{\pm0.05}$ |
| | Acc(C) | $98.42_{\pm0.72}$ | $96.42_{\pm1.27}$ | $98.37_{\pm0.68}$ | $92.02_{\pm2.53}$ |

Table 4: Data efficiency: Reduced number of negative samples. We vary the number of negative samples per batch from 1024 to 16384. The 16384 setting replicates the results of table 1. We report mean and standard deviation over 5 runs for $n = 3$.

| Group Type | Neg. sample batch size / Metric | 1024 | 2048 | 4096 | 8192 | 16384 |
|---|---|---|---|---|---|---|
| SO(n) | R²(x) | $99.51_{\pm0.26}$ | $99.66_{\pm0.09}$ | $99.48_{\pm0.71}$ | $99.77_{\pm0.04}$ | $99.81_{\pm0.04}$ |
| | R²(G) | $99.28_{\pm0.38}$ | $99.52_{\pm0.11}$ | $99.30_{\pm0.91}$ | $99.67_{\pm0.04}$ | $99.75_{\pm0.04}$ |
| | Acc(C) | $98.23_{\pm1.38}$ | $98.78_{\pm0.76}$ | $98.60_{\pm1.32}$ | $99.13_{\pm0.69}$ | $99.19_{\pm0.69}$ |
| O(n) | R²(x) | $99.55_{\pm0.07}$ | $99.63_{\pm0.11}$ | $99.62_{\pm0.28}$ | $99.75_{\pm0.10}$ | $99.80_{\pm0.06}$ |
| | R²(G) | $99.37_{\pm0.10}$ | $99.49_{\pm0.13}$ | $99.49_{\pm0.36}$ | $99.64_{\pm0.14}$ | $99.73_{\pm0.06}$ |
| | Acc(C) | $98.49_{\pm0.81}$ | $98.90_{\pm0.59}$ | $98.97_{\pm0.57}$ | $99.09_{\pm0.62}$ | $99.06_{\pm0.87}$ |
| GL(n) | R²(x) | $99.70_{\pm0.04}$ | $99.72_{\pm0.06}$ | $99.76_{\pm0.06}$ | $99.79_{\pm0.05}$ | $99.83_{\pm0.02}$ |
| | R²(G) | $99.53_{\pm0.07}$ | $99.56_{\pm0.08}$ | $99.61_{\pm0.10}$ | $99.65_{\pm0.08}$ | $99.72_{\pm0.05}$ |
| | Acc(C) | $98.92_{\pm0.53}$ | $99.09_{\pm0.58}$ | $99.11_{\pm0.53}$ | $98.76_{\pm0.63}$ | $98.42_{\pm0.72}$ |

**Discussion (Negatives):** As shown in Table 4, reducing the number of negative samples has a minimal effect on performance. While there is a slight, consistent downward trend in the equivariant metrics ($R^2(x)$, $R^2(G)$) as negatives decrease, the changes are very small and performance remains high ($> 99.2\%$) even with 16x fewer negatives.

Table 5: Data efficiency: Reduced number of samples per action ($k$) for fitting $\hat{R}(g)$. We vary $k$ from the theoretical minimum $n = 3$ up to $3n = 9$. Table 1 uses $k = 12$ ($4n$). We report mean and standard deviation over 5 runs for $n = 3$.

| Group Type | Samples per Action / Metric | 3 | 4 | 5 | 6 | 7 | 8 | 9 |
|---|---|---|---|---|---|---|---|---|
| SO(n) | R²(x) | $99.71_{\pm0.06}$ | $6.86_{\pm7.87}$ | $89.12_{\pm32.03}$ | $99.85_{\pm0.01}$ | $99.83_{\pm0.06}$ | $99.83_{\pm0.04}$ | $99.77_{\pm0.11}$ |
| | R²(G) | $-8.77_{\pm8.56}$ | $-9.59_{\pm28.67}$ | $88.16_{\pm33.06}$ | $99.63_{\pm0.05}$ | $99.66_{\pm0.13}$ | $99.70_{\pm0.07}$ | $99.66_{\pm0.16}$ |
| | Acc(C) | $99.20_{\pm0.52}$ | $51.17_{\pm19.08}$ | $97.47_{\pm4.53}$ | $99.40_{\pm0.47}$ | $99.30_{\pm0.74}$ | $99.07_{\pm0.94}$ | $99.10_{\pm0.68}$ |
| O(n) | R²(x) | $99.65_{\pm0.10}$ | $4.68_{\pm5.56}$ | $99.77_{\pm0.09}$ | $99.84_{\pm0.03}$ | $99.83_{\pm0.05}$ | $99.23_{\pm1.60}$ | $99.39_{\pm1.21}$ |
| | R²(G) | $-118.25_{\pm308.67}$ | $-0.03_{\pm0.09}$ | $99.09_{\pm0.35}$ | $99.59_{\pm0.09}$ | $99.67_{\pm0.10}$ | $98.75_{\pm2.56}$ | $98.95_{\pm2.15}$ |
| | Acc(C) | $99.00_{\pm0.82}$ | $54.32_{\pm12.03}$ | $98.88_{\pm0.81}$ | $99.18_{\pm0.50}$ | $99.13_{\pm0.90}$ | $97.58_{\pm4.59}$ | $97.94_{\pm3.98}$ |
| GL(n) | R²(x) | $98.46_{\pm1.13}$ | $5.56_{\pm9.94}$ | $89.63_{\pm28.86}$ | $99.63_{\pm0.03}$ | $99.76_{\pm0.09}$ | $99.80_{\pm0.04}$ | $99.80_{\pm0.07}$ |
| | R²(G) | $-0.28_{\pm0.47}$ | $-0.07_{\pm0.26}$ | $77.00_{\pm28.83}$ | $93.25_{\pm1.49}$ | $98.93_{\pm0.92}$ | $99.48_{\pm0.20}$ | $99.56_{\pm0.18}$ |
| | Acc(C) | $99.11_{\pm0.32}$ | $35.38_{\pm26.81}$ | $98.36_{\pm0.38}$ | $98.70_{\pm0.57}$ | $98.42_{\pm0.81}$ | $98.56_{\pm0.82}$ | $98.37_{\pm0.51}$ |

**Discussion (Samples per Action):** Table 5 confirms that the theoretical minimum of $k = n = 3$ samples is insufficient in practice. This is expected, as $k = n$ samples must form a full-rank system in latent space to uniquely identify $\hat{R}(g)$, which is unlikely to occur consistently. Using $k = 4$ also leads to instability. However, performance rapidly recovers. With $k = 6$ ($2n$), $R^2(G)$ is $> 99\%$ for $SO(3)/O(3)$ and $> 93\%$ for $GL(3)$. Using $k = 9$ ($3n$) yields near-perfect identifiability ($> 99.5\%$) for $R^2(G)$ across all groups, demonstrating practical applicability with a modest number of samples.

## C.3 Robustness to noise

To study the effect of noise, we adjust our data-generating process. We introduce a Gaussian noise term $\epsilon \sim N(0, \sigma^2)$ to the group action relationship in the latent space: $x' = R(g_i)x + \epsilon$. We test this for $GL(3)$, varying the noise standard deviation $\sigma$ from 0.0 to 0.1 and the number of samples per action ($k$) from 3 to 12. All other parameters follow the setup of Table 1.

Table 6: Robustness to noise for $GL(3)$. We add Gaussian noise $\epsilon \sim N(0, \sigma^2)$ to the latent transformation $x' = R(g)x + \epsilon$. We vary the noise standard deviation $\sigma$ and the number of samples per action $k$ ($3 = n$, $6 = 2n$, $9 = 3n$, $12 = 4n$) used to fit $\hat{R}(g)$. We report mean and standard deviation over 5 runs.

| Noise Std. | Group Type Metric Samples/Action | R²(x) | R²(G) | GL(n) Acc(C) |
|---|---|---|---|---|
| 0.0e+00 | 3 | $98.46_{\pm1.13}$ | $-0.28_{\pm0.47}$ | $99.11_{\pm0.32}$ |
| | 6 | $99.63_{\pm0.03}$ | $93.25_{\pm1.49}$ | $98.70_{\pm0.57}$ |
| | 9 | $99.80_{\pm0.07}$ | $99.56_{\pm0.18}$ | $98.37_{\pm0.51}$ |
| | 12 | $99.83_{\pm0.02}$ | $99.72_{\pm0.05}$ | $98.42_{\pm0.72}$ |
| 1.0e-05 | 3 | $97.89_{\pm1.88}$ | $-0.30_{\pm0.64}$ | $98.89_{\pm0.36}$ |
| | 6 | $99.62_{\pm0.06}$ | $93.24_{\pm1.20}$ | $98.68_{\pm0.78}$ |
| | 9 | $99.82_{\pm0.04}$ | $99.65_{\pm0.07}$ | $98.60_{\pm0.64}$ |
| | 12 | $99.82_{\pm0.03}$ | $99.70_{\pm0.06}$ | $98.48_{\pm0.61}$ |
| 1.0e-04 | 3 | $98.14_{\pm1.17}$ | $-28.47_{\pm65.56}$ | $99.00_{\pm0.36}$ |
| | 6 | $99.64_{\pm0.07}$ | $93.78_{\pm2.07}$ | $98.73_{\pm0.68}$ |
| | 9 | $99.83_{\pm0.02}$ | $99.65_{\pm0.06}$ | $98.45_{\pm0.47}$ |
| | 12 | $99.83_{\pm0.04}$ | $99.71_{\pm0.07}$ | $98.60_{\pm0.60}$ |
| 1.0e-03 | 3 | $98.58_{\pm0.28}$ | $-50.29_{\pm127.95}$ | $98.98_{\pm0.51}$ |
| | 6 | $99.58_{\pm0.11}$ | $93.33_{\pm1.76}$ | $98.51_{\pm0.67}$ |
| | 9 | $99.81_{\pm0.05}$ | $99.61_{\pm0.09}$ | $98.58_{\pm0.48}$ |
| | 12 | $99.81_{\pm0.04}$ | $99.68_{\pm0.08}$ | $98.50_{\pm0.70}$ |
| 1.0e-02 | 3 | $88.58_{\pm30.63}$ | $-1.02_{\pm1.98}$ | $98.98_{\pm0.50}$ |
| | 6 | $99.58_{\pm0.05}$ | $93.05_{\pm1.27}$ | $98.61_{\pm0.70}$ |
| | 9 | $99.81_{\pm0.04}$ | $99.59_{\pm0.08}$ | $98.46_{\pm0.81}$ |
| | 12 | $99.83_{\pm0.04}$ | $99.69_{\pm0.07}$ | $98.54_{\pm0.85}$ |
| 1.0e-01 | 3 | $94.56_{\pm11.34}$ | $-27.93_{\pm55.46}$ | $98.97_{\pm0.37}$ |
| | 6 | $99.58_{\pm0.06}$ | $91.76_{\pm0.23}$ | $99.04_{\pm0.56}$ |
| | 9 | $99.77_{\pm0.04}$ | $96.98_{\pm0.24}$ | $99.05_{\pm0.40}$ |
| | 12 | $99.82_{\pm0.03}$ | $97.98_{\pm0.13}$ | $98.94_{\pm0.74}$ |

**Discussion (Noise):** The results in Table 6 show that the $R^2(x)$ and $Acc(C)$ metrics are highly robust to noise, remaining stable even at $\sigma = 0.1$. As expected, noise primarily affects the identifiability of the group representation ($R^2(G)$). This effect is strongly dependent on the number of samples ($k$) used for the least-squares fit. With $k = 6$ ($2n$), $R^2(G)$ drops from 93.25% (no noise) to 91.76% ($\sigma = 0.1$). However, increasing the samples completely mitigates this. With $k = 9$ ($3n$), $R^2(G)$ only drops from 99.56% to 96.98%. With $k = 12$ ($4n$), the performance remains excellent, dropping from 99.72% to 97.98%. This demonstrates that the noise in the least-squares problem can be effectively averaged out by using more sample pairs, confirming the method's robustness.

## C.4    Infinite dSprites

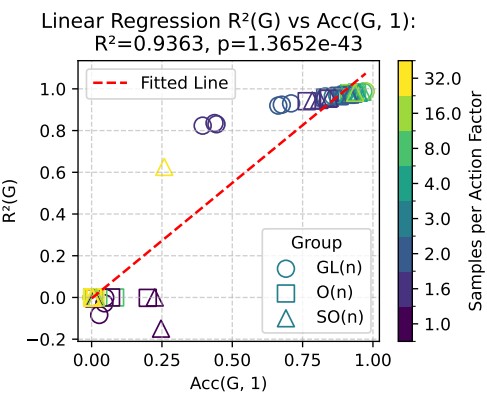 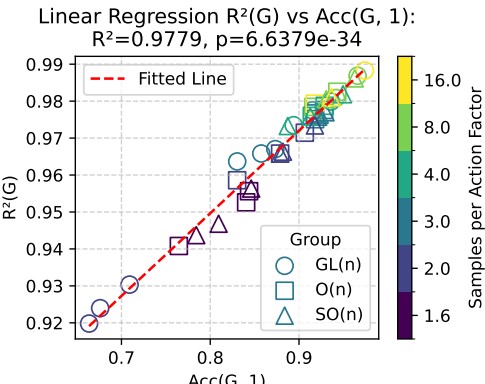

Figure 8: **Acc($G, 1$) as observable proxy for $R^2(G)$.** Comparison of our Top-1 NN-Lookup $Acc(G, 1)$ and group action identifiability metric $R^2(G)$ across EbC models trained on synthetic group datasets with varying samples per action factor $k$. Left: Results for the selected dataset configurations ($n = 5$, $d = 5$, $|C| = 1000$). Right: Filtered results for $Acc(G, 1) > 0.6$.

**The Acc(G) metric is suitable for practical hyperparameter selection.** In the main paper, we described the R2(G) metric to measure the quality of the group representation with respect to the ground truth group representation and latent factors. On practical experiments, this metric is not observable. Instead, on infinite dSprites but also for hyperparameter selection on the toy dataset, we consider the Acc(G, n) metric which performs a kNN lookup only in the reconstructed embedding space. Intuitively, this metric uses the computed embedding, and is a measure of self-consistency of the representation. As a validation for this choice, we varied the number of samples per action (one hyperparameter in the fitting procedure) and visualize the correleation between the observable Acc(G, 1) metric and the unobservable, but desired, R2(G) metric in Figure 8. On the full hyperparameter sweep, the metrics are significantly correlated with R2=93.6 (p<1e-10; Figure 8 left). When treating unsuccessful hyperparameter configurations as outliers, i.e., discarding models with Acc(G, 1) < 0.6, this correlation gets even more clear at R2=97.8 (p<1e-10; Figure 8 right).

**Hyperparameter selection indicates suitable group dimension on idSprites** We now leverage the $Acc(G, \cdot)$ metric to perform hyperparameter selection on the idSprites dataset. To provide a full picture, Figure 9 shows 10 variants of the metrics with differently strict criteria, considering top-1 to top-10 accuracies during the NN-lookup.

For 18 and 144 classes, we observe a clear increase in performance in all metrics at d=4 for the least conservative $Acc(G, 10)$ metric, however, the most conservative $Acc(G, 1)$ metric still indicates subpar performance at this dimensionality (below 50%). All metrics start saturating around a dimensionality of d=6, which would allow an embedding of $R_m \times Z_n \times Z_n$ into 3 circular dimensions.

Figure 10 shows example transitions at this hyperparameter point, which show close-to-perfect recovery of the group structure. However, as we overparameterize the model in terms of the group dimensions, the $Acc(G, \cdot)$ metrics improve slightly while the qualitative evaluation shows a noticeable decline in embedding and representation quality (Figure 11). In contrast to the results from Figure 8, this suggests a mismatch of the $Acc(G, \cdot)$ metric and the $R^2(G)$ metric which is unknown for this dataset.

We hypothesize that this is because the $Acc(G, \cdot)$ metrics a) are not able to measure the group action prediction exclusively, but instead also measure misclassification of the content, and b) don't measure the degree of the prediction error. Instead any type of classification error is counted with the same weight, no matter if the prediction was off by a small or large amount.

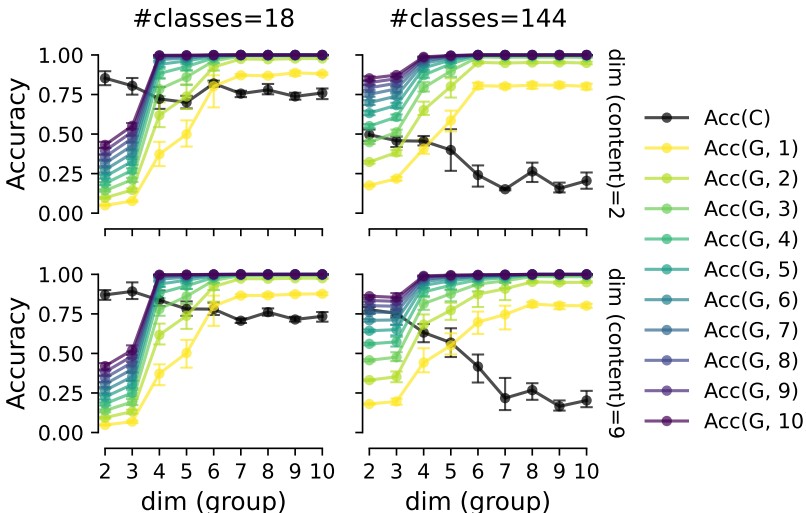

Figure 9: **Embedding Dimension Hyperparameter Sweep for idSprites dataset.** Comparison of EbC for the idSprites dataset across embedding dimensions for the group $n \in [2, .., 10]$ and for the content $d \in [2, 9]$ showing the observable Top-K NN-lookup metric $Acc(G, k)$ for $k \in [1, .., 10]$ as well as unobservable content classification accuracy $Acc(C)$. We show the comparison across two configurations of the idSprites dataset. Left: original dSprites configuration, Right: $16\times$ #shapes, $0.5\times$ #scales #rotations #translationsX #translationsY

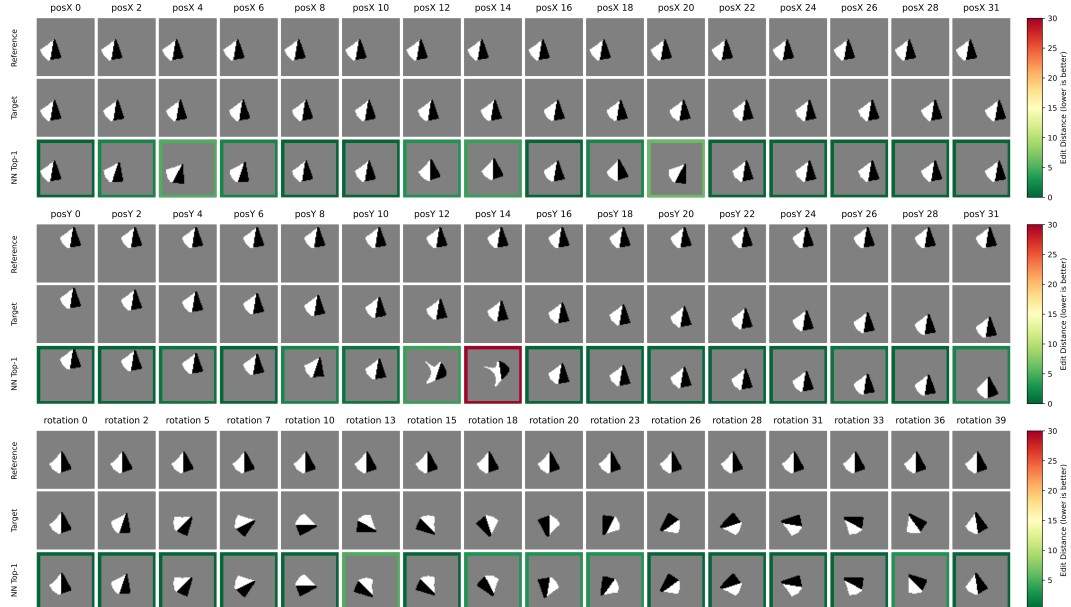

Figure 10: **Top-1 Nearest Neighbor Lookup based on EbC with 6D group dimensions.** For the idSprites dataset configuration with $|C| = 18$, we sample a random reference image $y$ and generate a sequence of transformed images $y'$ (targets) according to the different types of possible group actions. We estimate $\hat{R}$ from the embeddings, compute $\hat{R}x$ and perform a top-1 NN lookup across *all* embeddings to get a prediction in image space. In the three blocks of images we show action traces across posX translation (top), posY translations (middle), and rotations (bottom). Within each block we show the reference image (first row), the target images (second row), the top-1 NN image prediction (third row). We color the images in the third row according to the Manhattan distance between the associated true latent factors of the target and predicted image.

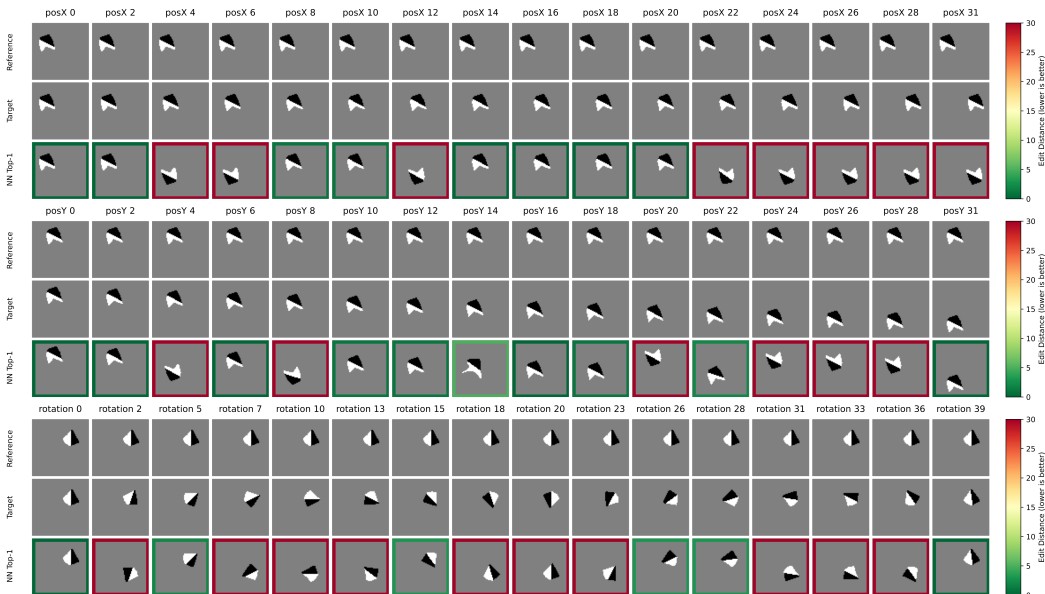

Figure 11: **Top-1 Nearest Neighbor Lookup based on EbC with 8D group dimensions.** Just like Figure 10 but for EbC with $n = 8$.

## C.5 Application to real-world data

Here, we demonstrate how EbC can be used on a real-world time series dataset to find group equivariant embedding spaces. We apply EbC to the rat hippocampus dataset recorded by Grosmark and Buzsáki [14]. We use the data as preprocessed by Zhou and Wei [55]. In this dataset, neural activity from CA1 neurons in hippocampus of multiple rats was recorded via implanted silicon probes. During the recordings of each session, the rats moved on a 1.6m long linear track. Neural activity, the position on the track, and the movement direction of the rat was recorded (see Figure 12A). We apply EbC to the data of the following three rats: "Achilles", "Buddy" and "Gatsby".

Using EbC for data analysis requires to specify how positive pairs are sampled and how positive pairs are grouped together into the same group action. This is conceptually close to CEBRA [43], where auxiliary variables are used to resample the dataset to encode particular neuroscientific hypotheses. As in CEBRA, we leverage the behavioral variables of position and movement direction, but discretize the position variable with a bin size of 1cm.

We propose three different sampling strategies, depicted in Figure 12B:

1. Group=Position: We consider all possible pairs of data points to be positive pairs and group positive pairs based on the difference in position regardless of the movement direction.

2. Group=Position & Direction: We consider all possible pairs data points to be positive pairs and group positive pairs based on the difference in position and difference in movement direction.

3. Group=Position, Content=Direction: We consider only those pairs of data points where the movement direction does not change and we group positive pairs based on the difference in position.

We compare our method to CEBRA-behavior [43] which is a state-of-the-art method on this dataset and closely related to EbC in the sense that is also an encoder-only representation learning method with identifiability guarantees. We compare both against CEBRA-Behavior using only the position variable and the variant using both position and movement direction as label information. We follow the same dataset split into train, validation and test used in [43], and leverage consistency across runs for selecting the best hyperparameters for each model. To match the "offset10-model" used by CEBRA as best as possible, we leverage a time-delay embedding for EbC with receptive field 10.

For the CEBRA models, we run a hyperparameter sweep over latent dimensions of $d \in \{6, 8, 12, 16, 32\}$ and train for 10k steps. For the remaining parameters, we use the default parameters specified via CEBRA's sklearn API. In particular, this means using the cosine distance in the loss (restricting embeddings to the hypersphere), using the "offset10-model" (CNN based model with embeddings normalized to the hypersphere), and "time offset"=10 specifying how positive pairs are related in terms of the time offset between them.

For EbC, we run a hyperparameter sweep across group (2 or 3) and content dimensions (0, 1, or 4), as well as over the number of samples per action $k$ for which we run experiments with ($k = 4d$, $k = 8d$). We use a three layer MLP with 128 hidden units per layer for the encoder. We compute the gradients through the linear regression modeling fitting as done in Appendix C.1. As for CEBRA, we train for 10k steps.

We compute the following three types of metrics:

1. **Decoding metrics**: We follow CEBRA's decoding setup using a KNN (with K=3) both for decoding the position and direction from the embeddings.

2. **Consistency metrics**: We use the consistency metrics across runs defined by CEBRA to measure how well different embedding spaces can be aligned to each other. This metric fits an affine transform to map from one embedding space to another and measures the $R^2$ score of the prediction. We use this on 5 runs of the same model with different seeds and compute the consistency between every resulting embedding space with every other embedding space of those runs.

3. **Linear Group Action (Position)**: To measure how well a given embedding space is linearly structured with regard to the position variable, we collect all embedding points related by a specific difference in the position variable (here: 5cm) and fit a linear model for which

we measure the $R^2$ score of the prediction. To make this metric fairer, we fit two separate models for pairs based on the direction label. The metric value we report is the average of these two individual $R^2$ scores.

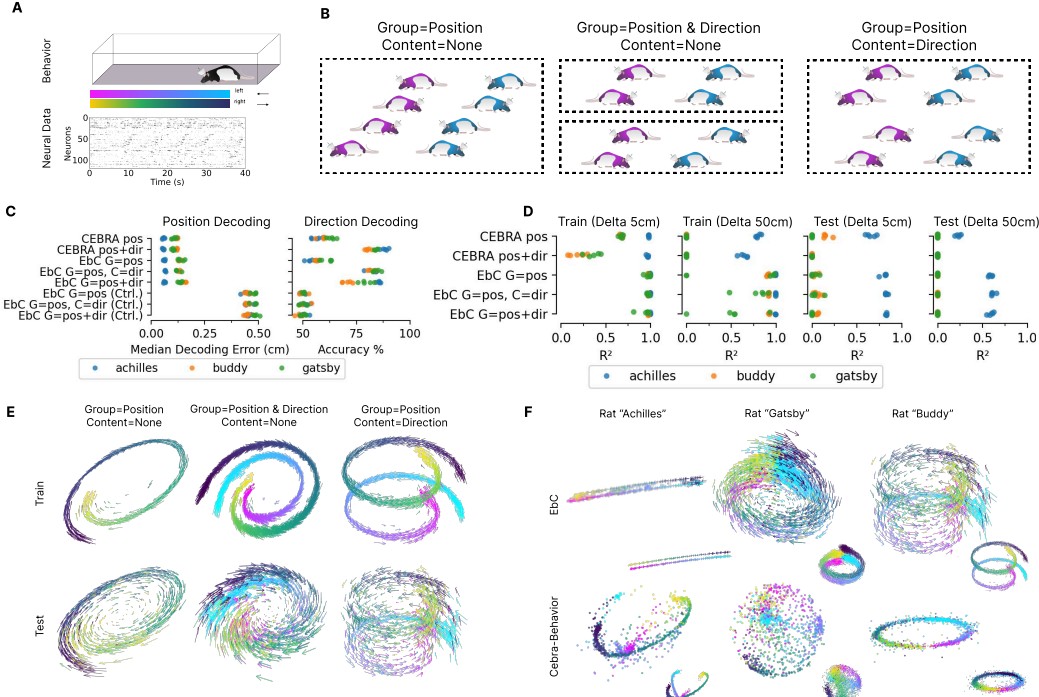

Figure 12: Application to neural time series data. (A) We used recordings from the hippocampus of rats running on a 1.6m long linear track. This yields two time series: neural activity $y_t$ and behavior (position $p_t$ and direction $d_t$). We use EbC to encode the neural activity with the behavior labels acting as labels determining the positive pairs and thereby the group structure. (B) Illustration of the different sampling strategies employed with EbC. (C) KNN-Decoding performance on the test set for decoding the position and direction label. We compare EbC to CEBRA and include a label shuffle control (D) Measure of linearity (for train and test set) of the embedding spaces conditioned on the difference in the position variable. Reporting the $R^2$ score of linear regression models trained on predicting the next point in latent space for pairs of data where the difference in position is 5cm. We also report the $R^2$ of using the same 5cm model on predicting 50cm ahead, i.e. 10 forward predictions in latent space. (E) EbC embeddings of the neural activity of the rat "Buddy" with different variations on how the behavior labels are used to determine positive pairs (F) EbC & CEBRA-Behavior Embeddings on rats Achilles, Gatsby and Buddy. Showing EbC model with group=position and content=direction, and CEBRA model with behavior labels position & direction.

**EbC reliably encodes position and direction.** Similar to CEBRA-behavior, EbC produces latent spaces from which both the position and movement direction of the rats can be decoded (Figure 12C). For position decoding, EbC and CEBRA-Behavior achieve comparable test performance: median errors of approximately 6 cm for "Achilles", ≈12–14 cm (EbC) and ≈11 cm (CEBRA-behavior) for "Gatsby", and ≈13–15 cm (EbC) and ≈12 cm (CEBRA-behavior) for "Buddy". In contrast, models trained with shuffled labels yield errors of ≈40–50 cm, confirming that both methods leverage meaningful neural–behavioral relationships for decoding.

For direction decoding, the EbC variant trained with direction as part of the content performs on par with CEBRA-behavior. As expected, the EbC model trained with both position and direction as group variables performs worse—particularly for "Buddy"—because this formulation enforces invariance to direction. Consistently, EbC models using only position as the group variable (no content) perform close to the shuffle baseline for direction, reflecting the intended invariance.

**EbC recovers latent linear group actions.** EbC reveals highly linear relationships in the latent space with respect to position (Figure 12D). At a local scale (5 cm position differences), the recovered embeddings exhibit near-perfect linear structure across all three rats. Notably, a linear model fitted only on 5 cm transitions generalizes well to a more global scale: predicting 50 cm ahead (10 steps) while retaining most of its predictive performance. In contrast, CEBRA-behavior does not consistently capture such linear group structure. At the 5 cm scale, performance is comparable for "Achilles" but weaker for "Buddy" and "Gatsby". At 50 cm, the linear trend largely disappears for "Buddy" and "Gatsby"; for "Achilles", the $R^2$ drops below 75%, whereas EbC remains at $\approx$80–100%. However, for both methods, linear models fitted on the train embeddings do not systematically generalize to the test embeddings, potentially due to the limited data size. For "Achilles" which has roughly twice as many neurons recorded as in other sessions, predictive performance generalizes to the test set for EbC.

**Sampling strategy induces structured representations.** As described above, EbC allows explicit control over the structure of the latent space through the sampling strategy defining positive pairs. For "Buddy", using **Group = Position** (direction ignored) yields embeddings that are equivariant to position and invariant to direction (Figure 12E). When **Group = Position+Direction**, the latent space organizes position as an approximately circular linear trajectory, while direction is encoded as orthogonal movement towards or away from the center of this "position circle". When grouping by position difference while holding direction fixed within a group (**Group = Position, Content = Direction**), EbC separates the two behavioral variables: direction is encoded along the content dimensions, while the group (style) dimensions capture the linear equivariant structure of position.

**Limitations and interpretation of model assumptions.** Both CEBRA-behavior and EbC provide identifiability guarantees (recovering the latent space up to a linear/affine transformation) under specific assumptions about the data-generating process. Violations of these assumptions break identifiability, and thus each method can be interpreted as testing a specific hypothesis on the structure of the neural–behavioral mapping. For CEBRA-behavior (Pos+Dir), nearby latent points correspond to samples that are close in time, position, and direction, with latent trajectories constrained to the hypersphere. For EbC, pairs of samples are related through a general linear group action in Euclidean space, where the group action is defined by the sampling strategy (as described above).

Three key differences follow: (1) CEBRA-behavior incorporates temporal proximity into the hypothesis; (2) CEBRA-behavior constrains samples to be close in latent space, whereas EbC enforces the existence of linear relations between groups of pairs; and (3) in our experiments, CEBRA-behavior restricts representations to the hypersphere, while EbC does not. In this sense, EbC imposes a stronger structural hypothesis on the data, particularly through the assumption of linear group actions.

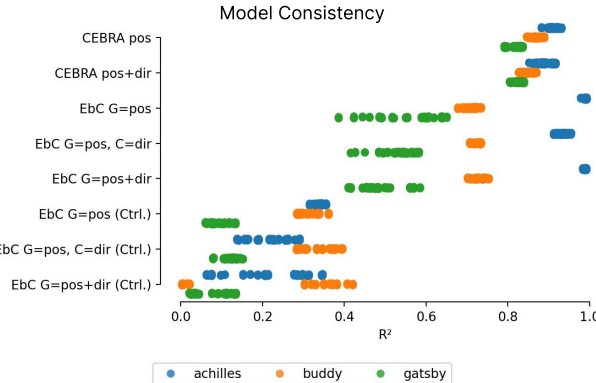

Figure 13: Consistency on the test set across model runs measured via the $R^2$ as defined by Schneider et al. [43].

**Consistency across runs.** Within this hypothesis-testing view, we assess whether embeddings from multiple runs (different random seeds) of the same model are linearly related, as predicted by the identifiability guarantees. For both EbC and CEBRA-behavior, we test whether a linear map exists that aligns embeddings across runs.

As shown in Figure 13, the highest consistency is observed for EbC (Group = Position and Group = Position+Direction) on "Achilles" (≈97–100%). For "Buddy", EbC reaches ≈70–75%, and for "Gatsby", ≈40–60%. In contrast, CEBRA-behavior achieves higher average consistency across rats: ≈85–93% (Achilles), ≈82–88% (Buddy), and ≈79–84% (Gatsby), although for Achilles, EbC exceeds CEBRA-behavior.

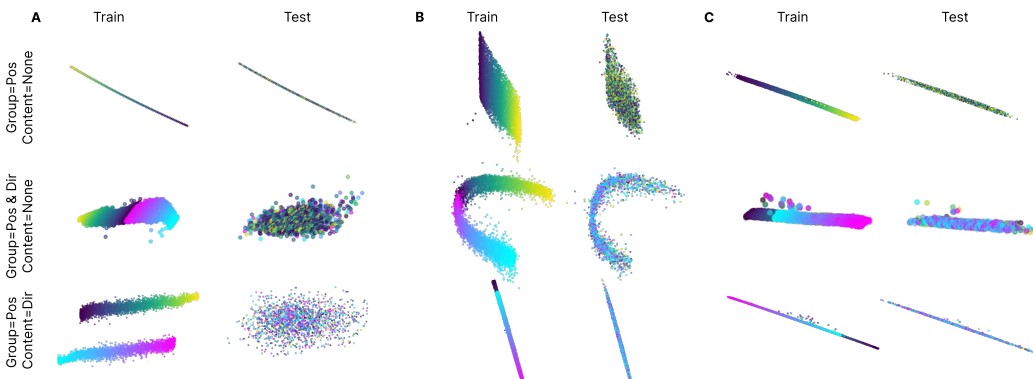

Figure 14: Embeddings of EbC label shuffle control experiments. Shows the same experimental setup as EbC experiments in Figure 12 but with the behavior labels randomly permuted before training.

**Shuffle control: importance of held-out evaluation.** The shuffled-label control highlights the necessity of train/(val)/test splits and cautions against interpreting structure from train embeddings alone. Even with randomized behavior labels, EbC imposes its specified group structure on the training latent space, producing structured embeddings across all rats (Figure 14). However, this structure differs from the non-shuffled embeddings in its general geometry and does not generalize to the test set when the neural–behavioral relationship is destroyed. When interpreting embeddings, it is hence important to consider that the mere linear arrangement of points in the embedding is consistently visible in the shuffled embeddings, but additional geometry in the non-shuffled embedding (e.g. alignment of positional information across directions; circular structure for some of the rats) is visible and non-trivial. On top, the observed structure in the non-shuffled embeddings transfers to the test set as an additional control to rule out the possibility of overfitting.

# D   Additional Literature Discussion and Related Work

Equivariant representation learning has been a prominent topic in machine learning research from different angles. Below we denote observed data as $y \in Y$, with $y' = g \cdot y$ as short-hand for the transformed data. Embeddings of the observed data are denoted as $x, x' \in X$, where $X$ is a vector space. Representations of the group are denoted as $R : G \to GL(n, X)$, and $x' = g \cdot x = R(g)x$. We focus on methods for learning vector embeddings $x$ of the data $y$, which are equivariant and/or invariant to group actions $g$. In contrast, orthogonal related work focuses on learning representations of the group given $X, Y$ [29, 50]. Another rich area of literature focuses on neural network architectures which are invariant or equivariant to specific predefined groups [3, 5, 10, 27, 42].

**Non-Invariant Methods.** Invariance to all possible group actions (augmentations) may hinder downstream tasks (e.g. color-invariance for flower-type classification). Xiao et al. [49] propose to use a contrastive learning framework to learn representation space in which every subspace is invariant to only one type of augmentation. Eastwood et al. [9] extend this by introducing a second entropy loss to encourage subspaces to become disentangled.

**Equivariant and Invariant Representation with known group action $g$ (explicitly observing g in a parameterized form).** Several encoder-based algorithms building on contrastive learning exist: E-SSL [4] embeds a reference sample of $y$ and multiple transformed samples $y_i' = g_i \cdot y$ into a latent space $x, x_i'$ which is split into an invariant and equivariant subspace. Invariant parts are learned via SimCLR [2] and equivariant parts through an auxiliary task that predicts the parametrized group action $g_i$ from $x_i'$. EquiMod [7] achieves equivariance by embedding pairs of $y, g \cdot y$ by explicitly modeling the group action in latent space via a neural network $u_\psi(x, g) = \hat{x}'$ and minimizing the distance between $\hat{x}'$ and $x'$. Park et al. [38] additionally parameterize the encoder and latent group action prediction model as G-equivariant neural networks. Garrido et al. [12] propose a variant using non-contrastive SSL losses.

Beyond contrastive learning, several approaches leverage (variational) autoencoders. Qi et al. [39] encode pairs of $y, y'$ into latent space and decode the group action $g$ from the concatenated embeddings $x, x'$. Jin et al. [23] embed $y$ into latent space, model the latent space group action as $x' = R(g)x$ and decode $y'$ from the predicted $x'$ assuming full knowledge of $R(g)$. Keurti et al. [24] use a linear prediction $x' = R(g)x$ where $R(g)$ is predicted by a neural network from the observed and parametrized $g$, essentially an auto-encoding variant of Garrido et al. [12].

**Equivariant & Invariant Representation with *unknown* group $g$ (implicitly observing $g$)** Although the aforementioned approaches share conceptual similarity in their goal of learning invariant and equivariant embeddings of data by modeling linear relations in latent space, EbC does not assume knowledge of the parametrized form of group actions $g$. Instead of observing information about $g$ directly, a second class of methods assumes that two or more pairs of data share the same underlying action, $y_i, g \cdot y_i$. This is a key assumption we also require for EbC.

Encoder-only methods include CARE [15], a contrastive learning framework to learn invariant an locally equivariant representations. CARE encodes two pairs of observations $(y_1, g \cdot y_1)$ and $(y_2, g \cdot y_2)$ with the same group action $g$ into a latent space such that the embeddings are related by the same matrix $R_g \in O(d)$ through $x_1' = R_g x_1$ and $x_2' = R_g x_2$. CARE achieves this by introducing an additional loss term to the InfoNCE loss, which maximizes the similarity between the dot products $x_1^T x_1'$ and $x_2^T x_2'$. Instead of applying this to a subspace of the embedding space, they apply both the invariant loss term of InfoNCE and their new equivariant loss term to the full embedding space and use a weighting factor to control the trade-off between invariant and equivariant representations. Yerxa et al. [51] propose a variation of CARE in which the embedding space is split into an invariant and equivariant subspace. Group actions are encoded as $x_1' = x_1 + b_g$ and $x_2' = x_2 + b_g$. STL [52] learns representations of (a) $y, y'$ such that $x, x'$ are equivariant to the group action $g$ and (b) the group action $g$ itself, again from two pairs of data. Similar to EquiMod [7], the latent group action is parameterized by a neural network. However, instead of assuming knowledge about $g$, they use a learned representation $R_g$ as the second input to the neural network: $x_1' = u_\psi(x_1, R_g)$ where $R_g$ is predicted by another network $R_\theta(x_2, x_2') = R_g$ from the latent representations of the second observed pair $(y_2, g \cdot y_2)$. To learn the correct representation of $R_g$, EquiMod is extended by a third loss term that maximizes the similarity of $R_\theta(x_1, x_1')$ and $R_\theta(x_2, x_2')$. Like CARE, they don't learn separate subspaces and instead use weighting factors for the invariant and equivariant loss terms to control the trade-off between invariant and equivariant representations in $X$.

The problem can also be approached with auto-encoding approaches: Winter et al. [48] learn embeddings of $y$ and $g$ in which the factorize $y' = g \cdot y$ into an representation $\hat{x}$ of $y$ which is invariant to $g$ and a representation $R_Y(g)$ that represents the action $g$ in the data space $Y$, such that $y' = R_Y(g)\delta(\hat{x})$ with $\delta$ being the decoder.

The Unsupervised Neural Fourier Transform (U-NFT) [28, 36] is an auto-encoder framework with a linear group action model in latent space. For their learning setup, Koyama et al. [28] assume access to multiple sequences of data points $\{y^{(i)} := (y_0^{(i)}, ..., y_T^{(i)})\}_{i=0}^N$ with $y_{t+1}^{(i)} = g_i \cdot y_t^{(i)}$, one sequence for each implicitly observed, but unknown group action $g_i \in G$. An autoencoder reconstructs $y_{t+1}^{(i)}$ from the predicted $\hat{x}_{t+1}^{(i)}$, which in turn is predicted from $\hat{x}_{t+1}^{(i)} = R(g_i)x_t^{(i)}$. The representation from these examples is estimated using least squares and a post-hoc basis transformation on the set of $R(g_i)$ to find a block diagonal representation $B(g_i)$ to facilitate disentanglement of the irreducible components of $R(g_i)$. Mitchel et al. [35] propose a variation of NFT, where the learning setup is restricted to directly produce a block-diagonal representation $R(g_i)$, avoiding the need for a post-hoc basis transformation. However, to achieve this, they also restrict $R(g_i)$ to be orthogonal.

Winter et al. [48] and Allingham et al. [1] only recover an invariant representation and model the group action separately in the data space. All other the methods in this section try to solve the same general task of learning equivariant and invariant representations of the data without explicit knowledge of the underlying group actions. CARE (Gupta et al., 2023) is the closest encoder-only (contrastive) approach to EbC, but constrains the embedding to the hypersphere, which imposes additional structure on the learned representation, while EbC can learn different topologies (e.g., torus in Fig. 4, hypersphere in Fig. 5). In terms of function and data requirements, U-NFT (Miyato et al., 2022; Koyama et al., 2023) is the conceptually closest work, but requires to learn a full generative model of the data.

