# OpenReview forum: "Equivariance by Contrast: Identifiable Equivariant Embeddings from Unlabeled Finite Group Actions"
_NeurIPS.cc/2025/Conference — NeurIPS 2025 poster_

### Official Review · Reviewer_s1fj · 2025-07-01

**Clarity:** 1
**Significance:** 1
**Originality:** 2
**Rating:** 2
**Confidence:** 3

**Summary:**

The paper presents a framework for learning embeddings that are equivariant with respect to the action of a finite group on the data. This is achieved by leveraging observed pairs consisting of a data point and its transformation under a fixed group element.

**Questions:**

The current state of the paper makes it very difficult to evaluate the work effectively. I understand the time and content constraints of the rebuttal phase; however, I believe it is both appropriate and necessary to request that the authors revise Section 2 - which outlines the proposed problem and methodology - as this revision is essential for enabling a meaningful re-evaluation of the paper.

It is important to note that this does not imply that only Section 2 requires revision. Rather, gaining a clearer understanding of the proposed method would facilitate a more informed and constructive assessment of the rest of the paper.

**Ethical Concerns:**

["NO or VERY MINOR ethics concerns only"]

**Final Justification:**

I remain concerned about the numerous typographical errors and the lack of clarity in several parts of the manuscript. Many of the clarifications provided during the rebuttal phase should be integrated directly into the paper to enhance its readability and comprehensiveness. These unresolved issues continue to detract from the overall presentation. Although I have revisited my evaluation in light of the improvements, my recommendation still leans toward rejection at this point.

**Limitations:**

.

**Paper Formatting Concerns:**

There is no issue with paper formatting.

**Quality:**

1

**Strengths And Weaknesses:**

**Strengths.** The supplementary material is provided and includes both the theoretical proofs and additional experimental details for the conducted studies. However, at this stage, I am unable to offer a substantive evaluation of the strengths of the work, due to the concerns outlined in the Weaknesses section below.

**Weaknesses.**

*1.* The paper contains numerous typographical errors, unclear notations, and ambiguous conceptual explanations, which significantly reduce the overall clarity and hinder the readability of the work. It is essential that the authors carefully address these issues in a revised version. Below, I list several examples from the first two sections - though this list is not exhaustive - and many more issues are readily apparent in the experimental sections.

- Line 5-7: The notation $R_m, \mathbb{Z}_n$ are not clear. What are periodic translations?
- Line 9-10: non-Abelian -> nonabelian or non-abelian
- Line 10: generalized linear group -> general linear group
- Line 18: allows to study -> allows us to study
- Line 19-22: Only when a group acts on a vector space (of finite dimensions), we have the group representation.
- The caption of Figure 2 is non-informative since it has too many typos in crucial equations. What does "y'_i = gy", or "x_i = R(g)x_i" mean?
- Line 21, 62, 64, etc.: \mapsto -> \rightarrow
- Line 34: questions -> question
- Line 38: perturbations[4] -> perturbations [4]
- Line 58-59: "which are grouped 59 into n + 1 pairs undergoing":  What does "pairs" mean here?
- Line 66: Information of  -> Information about
- Line 66: what is "u"? Is it "c"?
- Line 70: process Assume -> process. Assume
- Line 77: the second \phi(y_i') should be \phi'(y_i')
- Line 79: What is R(\Phi, \Phi')? The paper only defines R(g) or \hat{R}.
- Equation (3): What is S here?
- Line 84: where used -> where we use
- Line 97: What is Eq. (X)?
- Line 98: GL should not be in italic.
- Line 99: a -> an
- Line 112: Does "Assume that p_\phi = p_{id}" mean you assume that \phi is the identity map?
- Line 114: I am not sure why the original vector relates to the equation right after.

*2.* The discussion of related work is insufficiently thorough and lacks critical engagement with relevant prior literature.

*3.* The experiments are conducted solely on the dSprites dataset. Given that the authors reference a wide range of real-world applications - including biology, neuroscience, drug discovery, and computer vision - the paper should include at least one task or dataset from these domains to substantiate these claims.

---

> ### Author Rebuttal · Authors · 2025-07-31
>
> Dear reviewer, thanks a lot for your critical assessment of our work which raised several important points.
>
> **1. Typos and Questions.**
>
> > L 5-7, periodic translations:
> We mean translations with periodic boundaries.
>
> > Figure 2 caption; What does $y'_i = gy$, or $x_i = R(g)x_i$ mean?
>
> Apologies for the confusion, the new caption reads: $n+1$ samples are transformed using the same group action $g$, yielding samples $y$ and $y'$ in observation space. An encoder $\phi$ maps these samples into latent space ($\hat x, \hat x'$), where $n$ samples are used to estimate a linear representation $R$ of the action. The minimizer of the contrastive loss ensures that $\hat x_i = R \hat x_i$.
>
> > L 58-59: What does "pairs" mean here?
>
> Consider two data points with latent representation $(x,x')$ such that $x'=gx$; we observe $(f(x),f'(x))$. We leverage $n+1$ such pairs to jointly discover the latent space and a representation of the group. Note, this is directly defined in the definition following the sentence.
>
> > L 66: u/c
>
> Typo: we mean $c$ as used in Eq. 1 -- apologies.
>
> > L 79: What is $R(\Phi, \Phi')$?
>
> Typo: we mean $\hat R(\Phi, \Phi')$, as defined in Eq. 2.
>
> > Eq (3): What is $S$?
>
> Negative examples as implicitly described in L. 89; we added a sentence. These are uniformly drawn from the dataset.
>
> > L 97: Eq. (X)?
>
> Eq. 4
>
> > Line 112: Does "Assume that $p_\phi = p_{id}$" mean you assume that $\phi$ is the identity map?
>
> Yes, exactly, but this is a typo that slipped through from an earlier formulation. Given Eq 3, this should read "Assume that $p_{\phi} = p_{f^{-1}}$ [...]". Our goal is that if $\phi \circ f$ recovers a feature space in which we match the ground truth distribution, $\phi \circ f$ becomes linear (see Thm 1 in response to Reviewer bGkb).
>
> > Line 114: I am not sure why the original vector relates to the equation right after.
>
> The original vector $x$ is mapped to observed data $f(x)$ and then mapped back to a vector $\hat x = \phi(f(x))$. EbC assures that $\hat x = Lx$, even though $f$ can be an arbitrary non-linear function (as long as it is bijective).
>
> **To your question about Section 2**, we would be happy to discuss more specifics of the method. We believe that beyond the typos, the level of detail in Sec. 2 is self-sufficient. Could you let us know which part of the Methods need clarification?
>
> **2. Related work**
>
> We agree, and substantially extended our review of related work. See below.
>
> **3. Real-world data**
>
> Great suggestion. We added a real-world use case from neuroscience, where a rat is running on a linear track and hippocampal place cells are recorded. Please see our reply to reviewer zh3E.
>
> ### Extended discussion of related work
>
> Equivariant representation learning has been a prominent topic in ML research from different angles. Below we denote observed data as $y \in Y$, with $y'=g \circ y$ as short-hand for the transformed data. Embeddings of the data are denoted as $x, x' \in X$, where $X$ is a vector space. They are related via the representation $R: G \rightarrow GL(n, X)$ via $x' = g \cdot x = R(g) x$. We focus on methods for learning vector embeddings $x$ of the data $y$, which are equivariant and/or invariant to group actions $g$. In contrast, orthogonal related work focuses on learning representations of the group given $X$, $Y$ (Yang et al., 2023; Laird et al., 2024). Another rich area of literature focuses on neural network architectures which are invariant/ equivariant to specific predefined groups (Cohen et al., 2016; Satorras et al., 2021; Kondor et al., 2018; Finzi et al., 2020; Dehmamy et al., 2021).
>
> **Non-Invariant Methods.** Invariance to all possible group actions may hinder downstream tasks (e.g. color-invariance for flower-type classification). Xiao et al. (2021) propose to use a contrastive learning framework to learn invariant subspaces to different types of augmentation. Eastwood et al. (2023) extend this by introducing a second entropy loss to encourage disentangled subspaces.
>
> **Equivariant & Invariant Representation with known group action $g$ (explicitly observing g in a parameterized form).** E-SSL (Dangovski et al., 2021) uses contrastive learning to embed a reference sample of $y$ and multiple transformed samples $y_i'=g_i \circ y$ into a latent space $x$, $x_i'$, which is split into an invariant and equivariant subspace. Invariant parts are learned via SimCLR (Chen et al., 2020) and equivariant parts through an auxiliary task that predicts the parametrized group action $g_i$ from $x_i'$. EquiMod (Devillers et al., 2022) achieves equivariance by embedding pairs of $y$, $g \circ y$ by modeling the group action via a neural network $u_\psi(x, g) = \hat{x}'$ to approximate $x'$. Park et al. (2022) leverage G-equivariant neural networks. Garrido et al. (2023) propose a variant using non-contrastive SSL losses.
>
> Auto-encoder approaches form an alternative: Qi et al. (2022) encode pairs $y, y'$ into latent space and decode $g$ from concatenated embeddings $x, x'$. Jin et al. (2024) embed $y$ into latent space, model the group action as $x' = R(g) x$ and decode $y'$ from the predicted $x'$, assuming full knowledge of $R(g)$. Keurti et al. (2023) use a linear prediction $x' = R(g)x$ where $R(g)$ is predicted by a neural network from the observed and parametrized $g$, essentially an auto-encoding variant of Garrido et al., 2023.
>
> **Equivariant \& Invariant Representation with *unknown* group $g$ (implicitly observing $g$)** Although the aforementioned approaches share conceptual similarity in their goal of learning invariant and equivariant embeddings of data by modeling linear relations in latent space, EbC does not assume knowledge of the parametrized form of group actions $g$. Instead of observing information about $g$ directly, a second class of methods assumes that two or more pairs of data share the same underlying action, $y_i$, $g \circ y_i$. This is a key assumption we also require for EbC.
>
> Encoder-only methods include CARE (Gupta et al., 2023), a contrastive learning framework to learn invariant and locally equivariant representations. CARE encodes two pairs of observations $(y_1, g \circ y_1)$ and $(y_2, g \circ y_2)$ with the same action $g$ such that the embeddings are related by the same matrix $R_g \in O(d)$ via $x_1' = R_g x_1$ and $x_2' = R_g x_2$. CARE extends the InfoNCE loss with an equivariant term based on $x_1^T x_1'$ and $x_2^T x_2'$. The invariant loss term of InfoNCE and the new equivariant loss are weighted and applied to the full embedding space. Yerxa et al. (2024) propose a variation of CARE in which the embedding space is split into an invariant and equivariant subspace group actions are encoded as $x_1' = x_1 + b_g$ and $x_2' = x_2 + b_g$. STL (Yu et al., 2024), learns representations of $y, y'$ such that $x, x'$ are equivariant to the group action g and to learn a representation of $g$ itself, again from two pairs of data. Similarly to EquiMod, the group action is parameterized by a neural network. However, instead of assuming knowledge about g, they use a learned representation $R_g$ as the second input to the neural network: $x_1' = u_\psi(x_1, R_g)$ where $R_g$ is predicted by another network $R_\theta(x_2, x_2')=R_g$ from the latent representations of the second observed pair $(y_2, g \circ y_2)$. To learn the correct representation of $R_g$, EquiMod is extended by a third loss term that maximizes the similarity of $R_\theta(x_1, x_1')$ and $R_\theta(x_2, x_2')$. Like CARE, they do not learn separate subspaces and instead use weighting factors for the invariant and equivariant loss terms on a joint latent space.
>
> The problem can also be approached with auto-encoding approaches: Winter et al. (2022) learn embeddings of $y$ and $g$ in which the factorize $y'= g \circ y$ into an representation $\hat{x}$ of $y$ which is invariant to $g$ and a representation $R_Y(g)$ that represents the action $g$ in the data space $Y$, such that $y' = R_Y(g) \delta(\hat{x})$ with $\delta$ being the decoder.
>
> The Unsupervised Neural Fourier Transform (U-NFT; Koyama et al., 2023; Miyato et al., 2022) is an auto-encoder framework with a linear group action model in latent space.  For their learning setup, Koyama et al. assume access to N sequences of data points $(y^0, ..., y^i, ..., y^N)$ with $y^i = (y^i_0, ..., y^i_T)$   with $y^i_{t+1} = g_i \circ y^i_t$, one sequence for each implicitly observed, but unknown group action $g_i \in G$. An autoencoder reconstructs $y^i_{t+1}$ from the predicted $\hat{x}^i_{t+1}$, which in turn is predicted from $\hat{x}^i_{t+1} = R(g_i) x^i_{t}$. The representation from these examples is estimated using least squares and a post-hoc basis transformation on the set of $R(g_i)$ to find a block diagonal representation $B(g_i)$ to facilitate disentanglement of the irreducible components of $R(g_i)$. Mitchel et al. (2024) propose a variation of NFT, where the learning setup is restricted to directly produce a block-diagonal representation $R(g_i)$, avoiding the need for a post-hoc basis transformation. However, to achieve this, they also restrict $R(g_i)$ to be orthogonal.
>
> Winter et al. (2022) and Allingham et al. (2024) only recover an invariant representation and model the group action separately in the data space. All other the methods in this section try to solve the same general task of learning equivariant and invariant representations of the data without explicit knowledge of the underlying group actions. CARE (Gupta et al., 2023) is the closest encoder-only (contrastive) approach to EbC, but constrains the embedding to the hypersphere, which imposes additional structure on the learned representation, while EbC can learn different topologies (e.g., torus in Fig. 4, hypersphere in Fig. 5). In terms of function and data requirements, U-NFT (Miyato et al., 2022; Koyama et al., 2023) is the conceptually closest work, but requires to learn a full generative model of the data.

---

> > ### Comment · Reviewer_s1fj · 2025-08-05
> >
> > I thank the authors for their response. The revisions have improved the clarity of the paper, and the core contributions are now more accessible. Notably, the expanded discussion of related work provides valuable context and more clearly positions the paper within the existing literature.
> >
> > The inclusion of the new experiment based on a real-world neuroscience scenario—where hippocampal place cell activity is recorded as a rat traverses a linear track—adds empirical strength and highlights the practical relevance of the proposed method.
> >
> > Nonetheless, I remain concerned about the numerous typographical errors and the lack of clarity in several parts of the manuscript. Many of the clarifications provided during the rebuttal phase should be integrated directly into the paper to enhance its readability and comprehensiveness. These unresolved issues continue to detract from the overall presentation. Although I have revisited my evaluation in light of the improvements, my recommendation still leans toward rejection at this point.

---

> > > ### Author Response · Authors · 2025-08-06
> > >
> > > Dear reviewer,
> > >
> > > Thank you very much for following up and acknowledging the improvements through the provided clarifications, the expanded related work section, and the real world experiments. Of course, the content we posted during the rebuttal went into the updated manuscript.
> > >
> > > We apologize again about typos (which we clarified above), **but want to respectfully push back on your assessment that this prohibits understanding our method**, which is also mirrored in other reviews. From your suggestions on how to correct the flagged typos, it is apparent that the intended meaning was clear from the context given in the paper.
> > >
> > > What are you remaining concerns about Section 2 and other sections? We believe this section rather comprehensively outlines the steps for implementing EbC and its theoretical foundation, but **we would be happy to revise unclear parts of it and post them here**.

---

> ### Author Response · Authors · 2025-08-07
>
> Dear reviewer, in the meantime, we wanted to share **our revised Section 2**; most suggested changes were applied following the paragraph on "Implicit group representations". We hope that the revised section clarifies remaining concerns about our approach.
>
> Please note, two equations were too complex to render on OpenReview, marked with * below; in the paper, these equations are typeset in a single line as in our submission.
>
> ---
>
> **Implicit group representations.**
> We model the group representation via a non-parametric approach. Assume that for each group element $g\in G$, we are given two matrices $\mathbf{X},\mathbf{X}'\in\mathbb{R}^{M\times d}$, $M>d$, where the row vectors $(\mathbf{x}_i,\mathbf{x}_i')$ are related via the group element as $\mathbf{x}_i'=g \mathbf{x}_i$, $i\in\{1,\dots,M\}$. As a shorthand, we write $\mathbf{X}'=g\mathbf{X}$. Then, the expression
>
> $$
> \hat{\mathbf{R}}(\mathbf{X},\mathbf{X}')
> =\min_{\mathbf{R}\in\mathrm{GL}(d)}||\mathbf{X}'-\mathbf{X} \mathbf{R}^\top||_F^2 \Leftrightarrow
> \hat{\mathbf{R}}(\mathbf{X},\mathbf{X}')
> =(\mathbf{X}^\top\mathbf{X})^{-1}(\mathbf{X}^\top\mathbf{X}')
> $$
>
> is a representation of $G$ with $\mathbf{R}(g)=\hat{\mathbf{R}}(\mathbf{X},g\mathbf{X})$ for each $g\in G$. In practice, we do not have access to $(\mathbf{X},\mathbf{X}')$ directly, but only to a nonlinear projection of these points via a mixing function $\mathbf{f}$, denoted $(\mathbf{f}(\mathbf{X}),\mathbf{f}(\mathbf{X}'))=(\mathbf{Y},\mathbf{Y}')$. We map these observed samples to a feature space using a learnable encoder $\phi:\mathbb{R}^D\to\mathbb{R}^d$ and insert the resulting matrices into the above. Our goal is to optimize $\phi$ such that $\hat{\mathbf{R}}\bigl(\phi(\mathbf{Y}),\phi(\mathbf{Y}')\bigr)$ becomes a representation of $G$. An advantage of this approach is that both the feature space and the group representation are fully defined via the feature encoder $\phi$.
>
> **Equivariance by Contrast.** Given this structure, we propose our model, called Equivariance by Contrast (EbC). The intuition is depicted in Fig. 3: the model gets access to two observed sample sets related by the group action ($\mathbf{Y}$ and $\mathbf{Y}'$), along with a query sample $\mathbf{y}$. The objective is to infer the group action from $(\mathbf{Y},\mathbf{Y}')$, apply it to the query, and select the correct answer $\mathbf{y}'$ among a set of options $S$ that include $\mathbf{y}'$ and negative samples randomly drawn from the dataset.
>
> Formally, this objective is encoded via the likelihood*
>
> $$
> p_\phi(\mathbf{y}' | \mathbf{y},\mathbf{Y},\mathbf{Y}',S)
> $$
>
> $$
> = \exp(-||\mathbf{u}_\phi(\mathbf{y},\mathbf{Y},\mathbf{Y}')-\phi(\mathbf{y}'')||^2) /
> $$
>
> $$
> \sum_{\mathbf{y}''\in S}\exp(-||\mathbf{u}_\phi(\mathbf{y},\mathbf{Y},\mathbf{Y}')-\phi(\mathbf{y}'')||^2)
> $$
>
> We used the shorthand $\mathbf{u}_\phi$ to denote the operation of inferring the linear representation of the group element, $\hat{\mathbf{R}}(\phi(\mathbf{Y}), \phi(\mathbf{Y}'))$, and applying it to the feature vector of the query $\phi(\mathbf y)$:
>
>
> $$
> \mathbf{u}_\phi(\mathbf{y},\mathbf{Y},\mathbf{Y}')
> =\hat{\mathbf{R}}\bigl(\phi(\mathbf{Y}),\phi(\mathbf{Y}')\bigr) \phi(\mathbf{y})
> $$
>
> To find the optimal feature encoder $\phi$, we minimize the negative log-likelihood across all pairs of samples and uniformly sampled negative examples*,
>
> $$
> \min_{\phi}\mathcal{L}[\phi]
> $$
>
> $$
> =-E_{\mathbf{y},\mathbf{y}',\mathbf{Y},\mathbf{Y}',S}[\log p_\phi(\mathbf{y}'\mid\mathbf{y},\mathbf{Y},\mathbf{Y}',S)],
> $$
>
> which is closely related to the InfoNCE loss [30] augmented with the group-structure constraint.

---

### Official Review · Reviewer_bGkb · 2025-07-02

**Clarity:** 3
**Significance:** 2
**Originality:** 3
**Rating:** 5
**Confidence:** 4

**Summary:**

The authors present an algorithm to learn a representation of the group acting on a dataset as well as the latent space of the dataset. The author assume that the data $x_i $is only observable through a transformation $y_i = f(x_i)$. The authors address the problem by learning a mapping $\phi \approx f^{-1}$ (up to a linear transformation on x) and a matrix $R$ being a representation of the group action. They use a likelihood model as objective function to find $\phi$ and $R$. $\phi$ is represented with a neural network.

**Questions:**

- The model of Eq. (2) assumes that the same group action (i.e., the same matrix R) is applied to the complete dataset X. Why do the authors make this assumption? It would seem more realistic that different actions are applied on different data points.
- Why using the terms "content" and "style" in section 2?
- The authors do not detail how they deal with different (1000) group actions in their experiments. Indeed, the method on section 2 is only described in the case where a single group action is applied on the whole dataset.
- In the section "Metrics", the quantities of Eq. (8)-(10) use $h$, which should be unknown since $h=\phi\cdot f$  and $f$ is unknown. How are these quantities computed in practice?

**Ethical Concerns:**

["NO or VERY MINOR ethics concerns only"]

**Final Justification:**

I thank the authors for answering my concerns. I have raised my rating accordingly.

The authors have carefully and satisfyingly answered the points I made in my review.

The clarifications made by the authors in their nine responses should be included in the final version.

**Limitations:**

Yes.

**Paper Formatting Concerns:**

No.

**Quality:**

3

**Strengths And Weaknesses:**

# Strengths
- The introduction is well-written.
- The concepts of section 2 are introduced with enough clarity to be understood.

# Weaknesses
- The proof of Theorem 1 is given in Appendix A.3 while the main text presents an "informal" version if the theorem. Unfortunately, the appendix does not present a formal version of the theorem.
- The empirical evidence of Theorem 1 and Corollary 2 on a synthetic dataset, without noise, in section 4 is not convincingly explained.
- In a context where one seeks to learn the group acting on a dataset, it does not seem realistic to assume that both $Y$ and $Y'$ are known.
-In the subsection "Metrics", $\hat{h}$ is not defined.
- Experiments on $SO(n), O(n), GL(n)$, are limited to $n=3$.

---

> ### Author Rebuttal · Authors · 2025-07-31
>
> We thank the reviewer for their feedback. We appreciate the opportunity to clarify the points raised and answer open questions.
>
> ---
> ### Re: Formal Statement of Theorem 1
>
> You are correct that the manuscript would benefit from a more formal statement of the theorem in the appendix. We will update the appendix accordingly. Here is the full version we will include:
>
> **Theorem 1 (Identifiability of the Group Representation):**
> Let $\phi$ be a model parameterizing the distribution $p_\phi$ (Eq. 3), and let the model satisfy the diversity conditions in Def. 4. Let $f$ be an injective mixing function, and $g \in G$ be a group element according to Def. 3. Let $R$ be the representation of $G$, and $\hat R$ be the implicit representation of the group according to Eq (2) such that $\hat R(X, gX) = R(g)$ for pairs of transformed samples $(X, gX)$ and group actions $g \in G$.
>
> Then, for matching conditional distributions $p_\phi = p_{f^{-1}}$, the following holds for all points $x$ and group actions $g$ the dataset:
> 1. We recover the original vector space $\phi(f(x)) = L x$, up to an ambiguity $L \in GL(d)$.
> 2. We recover a representation of group in the dataset, $\hat R(f(X), f(gX)) = L R(g) L^{-1}$.
>
> ---
> ### Re: Empirical Evidence on Synthetic Data (Noise)
>
> We believe this is a potential misunderstanding. Our empirical validation exactly matches the assumptions of Theorem 1 and Corollary 2.
>
> To clarify the potential source of this misunderstanding. We indeed do require the assumption of the Diversity Condition (Appendix A, Def. 4). But the Diversity Condition doesn't not imply the presence of noise in the datasets. Instead it requires that there are enough group actions $g_i \in G$ with sufficient variation such that this leads to the required diversity in the set of positive and negative pairs.
>
> Nonetheless we do see the value in systematically studying the effect of noise on the data generating process.
>
> To facilitate this, we adjusted our data generating process to include a gaussian noise term in the group action relationship in latent space:
> $x' = R(g_i) x + \epsilon$ with $\epsilon \sim N(\mu=0, \sigma^2)$
> We set $\sigma^2 \in [0.0, 10^{-5}, 10^{-4}, 10^{-3}, 10^{-2}, 10^{-1}]$ and additionally run with a different amount of samples per action ($2n$, $3n$, $4n$, $5n$) for $GL(n)$ and $n=3$.
> The samples per action are the number of pairs $(x, x')$ we use to fit $\hat{R}(g_i)$ for any given $g_i$.
>
> Except for the described update to the group action relationship we follow the setup of Table 1 described in our paper.
>
> **Table 3.1: EbC on noisy data**
> |||Acc\(C\)|R²(G)|R²(x)|
> |-|-|-|-|-|
> |Samples Per Action|Noise Std.|||
> |6|0e+00|98.7±0.6|93.1±1.7|99.6±0.0|
> |6|1e-05|98.7±0.8|93.2±1.3|99.6±0.1|
> |6|1e-04|98.7±0.7|93.7±2.2|99.6±0.1|
> |6|1e-03|98.6±0.6|93.3±1.9|99.6±0.1|
> |6|1e-02|98.7±0.7|93.0±1.4|99.6±0.0|
> |6|1e-01|99.1±0.5|91.6±0.4|99.6±0.1|
> |9|0e+00|98.4±0.5|99.5±0.2|99.8±0.1|
> |9|1e-05|98.7±0.6|99.6±0.1|99.8±0.0|
> |9|1e-04|98.6±0.5|99.6±0.1|99.8±0.0|
> |9|1e-03|98.6±0.6|99.6±0.1|99.8±0.1|
> |9|1e-02|98.5±0.8|99.6±0.1|99.8±0.0|
> |9|1e-01|99.1±0.4|96.9±0.4|99.8±0.0|
> |12|0e+00|98.5±0.7|99.7±0.0|99.8±0.0|
> |12|1e-05|98.6±0.6|99.7±0.1|99.8±0.0|
> |12|1e-04|98.7±0.6|99.7±0.1|99.8±0.0|
> |12|1e-03|98.5±0.7|99.7±0.1|99.8±0.0|
> |12|1e-02|98.6±0.8|99.7±0.1|99.8±0.0|
> |12|1e-01|99.0±0.7|97.9±0.2|99.8±0.0|
> |15|0e+00|98.7±0.5|99.7±0.0|99.8±0.0|
> |15|1e-05|98.5±0.5|99.7±0.0|99.8±0.0|
> |15|1e-04|98.5±0.5|99.7±0.0|99.8±0.0|
> |15|1e-03|98.5±0.6|99.7±0.0|99.8±0.0|
> |15|1e-02|98.6±0.7|99.7±0.0|99.8±0.0|
> |15|1e-01|99.0±0.5|98.3±0.2|99.8±0.1
>
> **Summary:**
> Introducing Gaussian noise results in a drop in performance in terms of identifying an GL(n)-equivariant representation. This can be observed via the $R^2(G)$ metric, which reduces from ~93% to ~91% for EbC models with $2n$ sample pairs per action used for fitting $\hat{R}(G)$. But as we would expect for this noisy setting, increasing the number of samples per action seems to rectify this problem. As we increast to $3n$ samples per action, we get ~99.5% $R^2(G)$ for the noiseless setting and any noisy settings with $\sigma^2 <= 0.01$, while the largest amount of noise tested ($\sigma^2 <= 0.1$) results in ~96.9%.
> Finally, setting samples per action to $5n$, we get ~98.3% for the largest amount of noise tested and ~99.8% $R^2(G)$ for the less noisy settings.
>
> ---
> ### Re: Assumption of Known Data Pairs $(Y, Y')$
>
> We appreciate you raising this concern, which Reviewer 4zKK raised as well.
>
> We'd like to refer you to our answer there.
> But for your convenience, TLDR: This is a standard assumption all recent related work tackeling the same problem setting is making. See CARE (Gupta et al., 2023), STL (Yu et al., 2024), and U-NFT (Koyama et al., 2023). For a more detailed discussion on related work, we refer to our response to Reviewer s1fj.
>
> ---
> ### Re: Definition of $\hat{h}$
>
> We apologize for this oversight. What we mean here is $h$, not $\hat{h}$.
>
> ---
> ### Re: Experiments Limited to $n = 3$
>
> We'd like to refer you to the appendix where we already show results for $SO(n), O(n), GL(n)$ with $n \in [3,5,7,9]$.
>
> ---
> ### Re: Question on a Single Group Action
>
> We feel, there may be a missunderstanding. Of course we don't make the assumption that there exists only a single group action (=same matrix $R(g)$) that is applied to the complete dataset.
>
> To clarify:
> The observed variables are $Y, Y'$ not $X$. $X$ instead are the embeddings $X = \phi(Y)$.
> The data generating process used for the theory and empirically validation on synthethic data is defined via Eq.(1).
> The Eq.(2) on the other hand, simply describes how, given latent representations $X \in \mathbb{R}^{M \times d}$ and $X' \in \mathbb{R}^{M \times d}$ of $M$ observed sample pairs $Y, Y'$ we can compute an estimated representation $\hat{R}(X, X')=\hat{R}(g_i)$ of the group action $g_i$ these sample pairs share.
>
> Indeed for our experiments on the synthethic group dataset, we always generate 1000 random $R(g_i)$ from the respective group ($O(n), SO(n), GL(n)$).
> And in the case of dSprites, the $g_i = g_i^{(T)} \circ g_i^{(R)}$ observed during training are actually compositions of translations $g_i^{(T)}$ and rotations $g_i^{(R)}$ group actions.
>
> ---
> ### Re: Question on "Content" and "Style"
>
> With these terms we wanted to intutively refer to the unchanging components, the invariant part of the representations and the changing components so the equivariant part of the representations. Imagine the objects (content) of an image and the color, rotation, and scale (style) of these objects.
> However, if these terms are confusing or easily interpreted in a misleading way, we're happy to discuss alternatives.
>
> ---
> ### Re: Question on Dealing with 1000 Group Actions
>
> As referenced to earlier, the way we do this is defined via Eq.(2) and by extension Eq. (3-5).
> To clarify $Y, Y'$ or their embeddings $X, X'$ are **not** the whole dataset. Instead these represent a group of paired samples for which the pairs are all related by the same group action $g_i$. Think of them instead as $Y_i, Y_i'$.
> Note that Eq.(5) defines these are sampled from $p_{data}$, which represent the actual distribution over the full dataset.
>
> ---
> ### Re: Question on Computing Metrics
>
> Yes, this absolutely correct $f$ is unknown. However, $f(x)$ is not, because $f(x) = y$ represent the observed variable. See Eq. (1). On the other hand $x$ in this case are the ground truth latents, which usually are unknown. This is why we required a purely synthethic data setting to exactly verify the theoretical claims empirically.
>
> In practice for real-world datasets or benchmark datasets like DSprites where the ground truth latents are unknown these metrics can not be computed.
> This is why we introduce the Acc(G, k) and Acc(C, k) as proxy metrics.

---

> > ### Comment · Reviewer_bGkb · 2025-08-01
> > **Answer to authors**
> >
> > I thank the authors for answering my concerns. I have raised my rating accordingly.
> >
> > The authors have carefully and satisfyingly answered the points I made in my review.
> >
> > The clarifications made by the authors in their nine responses should be included in the final version.

---

> > > ### Author Response · Authors · 2025-08-09
> > >
> > > Thank you for your positive assessment following our discussion. We have incorporated the suggested changes into our paper and shared a summary as a global comment for all reviewers and the AC. We appreciate your time and suggestions.

---

### Official Review · Reviewer_4zKK · 2025-07-03

**Clarity:** 3
**Significance:** 2
**Originality:** 3
**Rating:** 4
**Confidence:** 3

**Summary:**

This paper considers the learning of equivariance by unsupervised/contrastive means.  The key mathematical result is on identifiability in this setting, by extending results of Roeder et al. a little bit.  The algorithmic approach follows the mathematical development and is demonstrated to work in one exemplary setting.

**Questions:**

Clarifying the differences between the present work and other past work on learning group-theoretic structure would be very helpful in clarifying the contributions here.  For example [Dehmamy et al., "Automatic symmetry discovery with lie algebra convolutional network," NeurIPS 2021], [Yang, et al., "Generative adversarial symmetry discovery," ICML 2023], or [Yu et al., "Information lattice learning," JAIR 2023].  See also references therein and thereto.  Can you do that?  It would help with your claim about "group representation learning from unlabeled observational data is feasible at all".

**Ethical Concerns:**

["NO or VERY MINOR ethics concerns only"]

**Final Justification:**

revised up, given the clarifications w.r.t. past literature

**Limitations:**

yes, they have addressed limitations.  no discussion of potential negative societal impact.

**Paper Formatting Concerns:**

formatting seems fine

**Quality:**

3

**Strengths And Weaknesses:**

STRONG: The problem of symmetry discovery is important in numerous settings, and the findings here for such a problem are intriguing.  Note that the setting is where there are batches of samples under the same transformation.  The paper hits all the points of formulation, theory, algorithm, and basic experimental demonstration.


WEAK: The seemingly non-native English phrasing, e.g. "Group theory allows to study this structure" -> "Group theory allows studying this structure" could be improved.

Even though Thm 1 is the main result, the technical diversity conditions are omitted in the main text: to this reviewer, this feels like a significant shortcoming that is easily rectified by moving them up from Appendix A.

The mathematical advance over Roeder et al [14] seems marginal.

I am not an expert in the motivating examples, but the batched data needed for this approach seems like a very strong assumption.

---

> ### Author Rebuttal · Authors · 2025-07-31
>
> We thank the reviewer for recognizing the importance of the problem and for their valuable feedback. We are happy to clarify the points raised.
>
> Regarding the English phrasing, we appreciate you pointing this out and will perform a thorough proofread for the camera-ready version.
>
> ---
> ### Re: Placement of Diversity Conditions
>
> Thank you for raising this concern, we agree that the theory section would benefit from highlighting the full set of assumptions more prominently and concisely. in the revised version of the main text.
>
> We would like to point out thought, that the diversity condition has been (informally) introduced in the main text (lines 107-111) right before stating the Theorem 1. We will move the consice but complete deftinition of all conditions from Appendix A into the main theory section to make Theorem 1 and its assumptions clearer.
>
>
> ---
> ### Re: Mathematical Advance Over Roeder et al. [14]
>
> Our Theorem 1 shows that group representations can be learned via the implicit represention in Eq. 2 and provides an identifiability proof for jointly learning the latent space *and* the group action using EbC. This is orthogonal to the key result in Roeder et al. [14].
>
> We believe that you refer to the fact that as part of our proof, we indeed leverage prior results from Roeder et al. [14], specifically we extend the canonical discriminative form towards Euclidean spaces (Theorem 5, see appendix) and use this form as part of our proof. This extension is indeed marginal, but a technical detail of our proof strategy.
>
> Does this clarify the concern? We would be happy to discuss more details.
>
> ---
> ### Re: Batched Data Assumption
>
> We appreciate the sentiment and are happy to elaborate on this:
>
> * **Common Assumption:** All of the recent related works known to us, which tackle this unsupervised problem, such as CARE (Gupta et al., 2023), STL (Yu et al., 2024), and U-NFT (Koyama et al., 2023), all require observing multiple pairs of data transformed by the same unknown action. Our data requirement is therefore in line with the current state of the art for this specific problem setting.
>
> * **Practical Scenarios:** This data structure naturally arises in many scientific and systems identification domains where one observes a system before and after an intervention/pertubation/transformation. For example:
>     * **Genomics (CRISPR Screens):** Measuring gene expression in cells before and after a specific gene knockout is applied to a batch of them.
>     * **Structural Biology:** Observing a protein's structure before and after a specific ligand binds.
>     * **Neuroscience (fMRI):** Recording brain activity at rest and then again after presenting a specific stimulus to a subject.
>
>
> For the most straight-forward application of EbC this requires discrete actions.
>
> However, as we demonstrate in our **new real-world experiment (see response to Reviewer zh3E)**, our method can be easily adapted to work effectively even with continuous auxiliary variables by discretizing the action space, showing its flexibility beyond these examples.
>
> ---
> ### Re: Clarifying Contributions vs. Other Past Work
>
> Thank you for suggesting these references. They allow us to better position our work. For a broader  and more detailed comparison, we **highly recommend** checking out **our extended related work discussion in the response to Reviewer s1fj**.
>
> * **Dehmamy et al. (NeurIPS 2021):** This work on Lie algebra convolutional networks is an excellent example of building **equivariant network architectures**. This line of work is complementary to ours. Such methods use prior knowledge of a symmetry group to build a specific inductive bias into the model. In contrast, our work aims to **discover** the unknown symmetry transformation from data without pre-specifying the network architecture to be equivariant to a specific type of groups.
>
> * **Yang et al. (ICML 2023):** This work also learns a matrix representation $R(g)$ of a group action. However, a key difference is that their representation acts directly on the *observed data* (i.e., $y' = R(g)y$). Our method solves a different problem: learning a latent representation $\phi(y)$ and a group action that applies in that *latent space*. This allows our method to handle complex, high-dimensional observations where the corresponding group action in data space may not be linear.
>
> * **Yu et al. (JAIR 2023):** Thank you for this reference. Our understanding of "Information Lattice Learning" is that it learns transformations between data points in the *data space*, without an explicit connection to group theory or the goal of recovering a linear group representation. The "information lattice" provides a different, non-algebraic structure for understanding these relationships. We therefore see the goals and technical approaches as distinct from ours.

---

> > ### Comment · Reviewer_4zKK · 2025-08-06
> >
> > Thanks to the authors for clarifying the standardness of the problem formulation of batched data, and how it arises in several practical scenarios.  Also for clarifying the mathematical advance over Roeder et al. in terms of implicit representations.  This, together with the neuroscience example added in response to other reviewers, gives me greater confidence in the work.
> >
> > If the paper is accepted, I would recommend including the longer discussion of other literature not just in the rebuttal but also in the manuscript itself, since it helps contextualize.  (As far as I know, the information lattice is very much group-theoretic and algebraic: in the finite group setting, it is equivalent to the subgroup lattice.  Please check.)

---

> > > ### Author Response · Authors · 2025-08-06
> > >
> > > Dear reviewer,
> > >
> > > Thank you for the follow-up. We're glad we could address your questions.
> > >
> > > To clarify, we updated our manuscript with the additional literature suggestions, both in the introduction to the paper, and as a comprehensive review in the supplementary material (due to space constraints). We believe that this positions EbC much better in the existing literature, and thank you again for your suggestions.
> > >
> > > If you have further questions and suggestions for improvements we can address, please let us know.

---

### Official Review · Reviewer_zh3E · 2025-07-10

**Clarity:** 3
**Significance:** 3
**Originality:** 3
**Rating:** 5
**Confidence:** 3

**Summary:**

The paper proposes a method for learning representation of a group from data through contrastive learning approach with theoretical guarantees and verifies on synthetic and 2D image datasets.

**Questions:**

- Figure 2: The encoder in Caption is f, and there is another encoder \phi after the samples, which is a bit confusing. What is Q in the Figure 2?
- I have less background in neuroscience and biology, and I’d appreciate some concrete examples on this.

**Ethical Concerns:**

["NO or VERY MINOR ethics concerns only"]

**Final Justification:**

I read the author response and other reviews. I would like to acknowledge the very detailed response of the authors to my concerns and questions. The response addresses most of my questions, and I think my comment on e.g., data efficiency experiment should not be expected to be fully addressed during the short rebuttal period. I still hold my view that this paper should be accepted to the conference and am happy to defend. I encourage the authors to incorporate the new materials and further strengthen in the paper revision.

**Limitations:**

Yes

**Quality:**

4

**Strengths And Weaknesses:**

**Strengths:**

- The paper is very well written, with clear motivation and exposition.
- Figure 3 provides an intuitive visual explanation of the proposed method, making the main idea easy to understand.
- Theoretical results (e.g., Theorem 1) demonstrate identifiability of the group action and latent space under mild assumptions, which is uncommon for self-supervised equivariant learning methods.
- The approach is simple and efficient, relying on contrastive learning with an encoder-only pipeline and on-the-fly least-squares fitting, avoiding the complexity of generative models.
- Strong empirical results: Experiments cover a diverse set of group structures (e.g., SO(3), O(3), GL(3), and product groups), demonstrating the generality of the method. Achieves near-perfect latent reconstruction and group prediction accuracy, outperforming relevant baselines on synthetic data.

**Weaknesses:**

- Data efficiency is not systematically studied; there is no analysis on how performance scales with dataset size, batch size, or the number of negative samples, which is especially relevant given the theoretical assumptions on required samples. I would suggest including experiments sweeping dataset size and batch/negative counts to measure convergence and data efficiency.
- Comparison to baselines is somewhat limited; recent related approaches such as Park et al. (ICML 2022), Homomorphism Auto-Encoder (Keurti et al., ICML 2023), and other group-equivariant learning methods are not included or discussed in detail.
- The key assumption (that all n+1 samples in a mini-batch share the same, unknown group element) is not tested or validated on real-world datasets; it is unclear how well the method handles weaker or approximate group structure in practical settings.
- Application to real-world data is limited; all experiments are on synthetic datasets (e.g., Infinite-dSprites), with no demonstration on realistic or noisy sequence data.

**Additional References Suggested:**

- Park et al., “Learning Symmetric Embeddings for Equivariant World Models,” ICML 2022.
- Keurti et al., “Homomorphism Auto-Encoder,” ICML 2023.
- Winter et al., “Learning Invariant and Equivariant Representations in Unsupervised Group Settings,” 2022.

---

> ### Author Rebuttal · Authors · 2025-07-31
>
> We would like to thank the reviewer for their positive assessment and very helpful and constructive feedback.
>
> Before addressing the specific points, we'd like to highlight that on synthetic group data our emprically results also extend beyond $n=3$ to $n \in [5,7,9]$, see Appendix C, Fig. 7.
>
> ### Re: Data efficiency
> We very much agree with your proposal. We have run new experiments to analyze data efficiency by varying dataset size, number of negative samples, and samples per action (for fitting $R(g)$), keeping all other parameters identical to our s in Table 1. We will add the full results to the appendix.
>
> Note: The following results also exist for SO(n), O(n), but GL(n) has performed strictly worse. Happy to provide SO(n), O(n) results upon request.
> Note 2: We have marked the hyperparameter setting corresponding to the replication of Table 1 in each of the following result tables with **
>
> **Table 1.1: EbC on reduced dataset size, $n=3$.**
> |       | Dataset Size    | 50k    | 100k   | 500k   | 1000k **   |
> |-|-|-|-|-|-|
> | **Group Type** | **Metric**              |  |  |  |   |
> | GL(n) | R²(x)              | 99.8±0.0 | 99.8±0.0 | 99.8±0.0 | 99.8±0.0  |
> | GL(n) | R²(G)              | 99.7±0.1 | 99.7±0.0 | 99.7±0.0 | 99.7±0.0  |
> | GL(n) | Acc\(C\)             | 91.9±0.9 | 97.2±0.8 | 98.5±0.6 | 98.5±0.7|
>
>
> **Summary:** The results show that our method is highly robust to smaller dataset sizes. Even with **20x less data**, the identifiability of the equivariant representation remains nearly perfect ($R^2(x)$ and $R^2(G)$ stay above 99.7%). The primary effect of smaller datasets is a moderate drop in accuracy for the invariant part of the embedding ($Acc(C)$).
>
> **Table 1.2: EbC on reduced number of negatives in a batch, $n=3$.**
> |       | Batch Sizes (Negatives)   | 1024     | 2048     | 4096     | 8192     | 16384**    |
> |-|-|-|-|-|-|-|
> | Group Type | Metric                         |  |  |  |
> | GL(n) | R²(x)                         | 99.7±0.0 | 99.7±0.1 | 99.8±0.1 | 99.8±0.1 | 99.8±0.0 |
> | GL(n) | R²(G)                         | 99.5±0.1 | 99.6±0.1 | 99.6±0.1 | 99.6±0.1 | 99.7±0.0 |
> | GL(n) | Acc\(C\)                        | 98.9±0.6 | 99.1±0.6 | 99.1±0.5 | 98.8±0.6 | 98.5±0.7 |
>
>
>
> **Summary:** Reducing the number of negative samples has a minimal effect on performance. While there is a slight downward trend in the metrics, the changes are very small and often close to the standard deviation across runs.
>
> **Table 1.3: EbC on reduced number of samples per action for fitting the linear least squares, $n=3$.**
> |       | Samples Per Action  | 3           | 4          | 5         | 6        | 7        | 8        | 9        | 12**       |
> |-|-|-|-|-|-|-|-|-|-|
> | Group Type | Metric                    | |  |
> | GL(n) | R²(x)                    | 98.5±1.1    | 5.5±10.0   | 89.6±28.8 | 99.6±0.0 | 99.8±0.1 | 99.8±0.0 | 99.8±0.1 | 99.8±0.0 |
> | GL(n) | R²(G)                    | -1.4±1.7    | -7.3±21.7  | 76.9±28.8 | 93.1±1.7 | 98.9±0.9 | 99.5±0.2 | 99.5±0.2 | 99.7±0.0 |
> | GL(n) | Acc\(C\)                   | 99.1±0.3    | 35.4±27.1  | 98.4±0.3  | 98.7±0.6 | 98.4±0.8 | 98.6±0.8 | 98.4±0.5 | 98.5±0.7 |
>
>
> **Summary:** These results confirm that the theoretical minimum of $n$ samples is insufficient in practice. This is expected, as achieving the theoretical limit would require the sampled pairs to form a full-rank system after being encoded into latent space, which is unlikely to occure consistently throughout the model training. However, performance recovers to $>99\%$ with a modest number of samples (e.g., >$2n$), demonstrating practical applicability.
>
> ### Re: Comparison to baselines
> We appreciate this point and have added an extended related work section to the paper (see the **full discussion in the response to reviewer s1fj**). The primary distinction between our method and the suggested works (**Park et al. '22**, **Keurti et al. '23**) is that **our method does not require ground-truth knowledge about group actions $g$**.
>
> * **Park et al. (ICML 2022):** This method requires known group actions $g$ as input to its latent action model and requires the network architecture itself to be G-equivariant, presupposing knowledge of G.
> * **Keurti et al. (ICML 2023):** This work uses an Autoencoder framework and also requires known parametrized $g$ to learn a matrix representation $\hat{R}(g)$. While its goal of finding a linear representation is similar, it relies on a stronger supervisory signal and the overhead of a decoder.
>
> In contrast, our approach only requires the weak assumption that sets of sample pairs underwent the *same unknown* transformation.
>
> ### Re: Weak or approximate group structure & **Validation on real-world dataset**
> We agree that these are crucial points and have addressed them both theoretically and empirically.
>
> **Theoretically**, our identifiability proof (see Appendix) does not rely on a strict group structure. The core assumption about the data generating process that is used in the proof is that pairs of latent points $(x, x')$ are related by a linear map, $x' = A_ix$. While we frame this in the context of group representations ($A_i = R(g_i)$), the proof does not require the set of matrices $\{A_i\}$ to form a group. This inherent flexibility suggests our method is suited for data with approximate or learned symmetries as well.
>
> To **empirically** validate this, we applied our method to a **real-world neuroscience dataset** (Grosmark & Buzsáki, Science 2016).
> **Data:** We used recordings from the hippocampus of rats running on a **1.6m** long linear track. This yields two time series: neural activity $\{y_t\}$ and behavior (position $p_t$ and direction $d_t$).
> **Setup:** We formed pairs $(y_t, y_{t+k})$ and clustered them based on the change in behavior $\Delta p_{t,k} = p_{t+k} - p_t$. Each cluster of pairs represents an approximate, unlabeled transformation of the same type (action). We then applied our method (EbC) to learn a behavior-equivariant latent space from the neural data. We compare against **CEBRA-Behavior** (Schneider et al., Nature 2023), a state-of-the-art method for this task, using the same data splits and evaluation protocol.
> **Evaluation:** We decode position (k-NN Regressor) and direction (Logistic Regression) from the learned embeddings.
>
> **Table 2.1: Position Decoding (Median Absolute Error in cm)**
> |Method|Rat Name|Train|Val|Test|
> |-|-|-|-|-|
> |CEBRA-Behavior|achilles|01.67±0.04|06.93±0.42|05.57±0.51|
> |CEBRA-Behavior|buddy|06.13±0.17|12.41±0.66|12.40±0.63|
> |CEBRA-Behavior|gatsby|06.68±0.14|11.65±1.09|11.40±0.30|
> |EbC|achilles|00.83±0.06|04.74±0.35|05.76±0.47|
> |EbC|buddy|01.24±0.10|14.04±0.70|16.02±1.55|
> |EbC|gatsby|00.89±0.02|08.76±0.24|14.28±0.87|
>
>
>
> **Table 2.2: Direction Decoding (Accuracy %)**
> |Method|RatName|Train|Val|Test|
> |-|-|-|-|-|
> |CEBRA-Behavior|achilles|60.15±0.32|55.95±0.98|59.62±0.79|
> |CEBRA-Behavior|buddy|60.26±0.25|69.30±0.81|58.33±1.36|
> |CEBRA-Behavior|gatsby|62.78±0.64|56.94±1.81|65.96±0.77|
> |EbC|achilles|92.17±0.66|88.62±1.77|82.20±0.50|
> |EbC|buddy|85.40±3.38|84.74±4.57|79.06±5.28|
> |EbC|gatsby|83.62±2.70|84.58±1.24|79.18±0.68|
>
>
> **Results:** Our method (EbC) performs close to / on par with CEBRA-Behavior for position decoding (test set relative error of 3.5-8.9% for EbC vs. 3.5-7.7% for CEBRA). Crucially, EbC **significantly outperforms the baseline in decoding the direction** of movement, achieving 79-82% accuracy on the test set compared to CEBRA's 58-69%. This successful application demonstrates that EbC can learn meaningful representations from noisy, real-world data with only approximate symmetries.
>
> Finally, we also ran a systematic study on the effect of noise within the data-generating process. We refer to our reply to bGkb for these results.
>
> ### **Answers to Questions**
>
> #### Q1 - Figure 2
> We apologize for the confusion caused by the typo and notation. We will correct Figure 2 in the revision. To clarify:
> * $f$: Represents the unknown data-generating function, an injective map from a true latent space $x$ to the observed samples $y$.
> * $\phi$: The **learnable encoder** that maps observed samples $y$ to an embedding space.
> * In the caption, it says "encoder $f$", which is a very unfortunate typo and should read "encoder $\phi$"
> * $Q$: This was a typo and should be $\hat{R}(g)$. It represents the **estimated linear group representation** that is computed on-the-fly via linear least-squares within each mini-batch.
>
> #### Q2 - Example applications
> We are happy to provide more concrete examples. Our framework is broadly applicable to any "intervention" setting where one observes a system's state before and after a common transformation is applied to a set of samples.
> * **Medicine:** One could study the effects of various drugs by observing patient physiology before and after treatment. The model could learn a "drug-effect-equivariant" representation, where the transformation corresponds to the physiological change induced by a specific drug.
> * **Cell Biology:** In single-cell RNA analysis, researchers measure gene counts, perform an intervention like a gene knockout, and measure again. Our method could identify representations equivariant to the effects of specific gene edits without prior knowledge of those effects.
> * **Time-Series with Auxilary variable:** As demonstrated in our new experiment, the framework can be creatively applied to time-series data where an auxiliary variable (like position) allows for grouping time-steps that represent a similar, unknown evolution of the system.

---

> > ### Author Response · Authors · 2025-08-06
> >
> > Dear reviewer,
> >
> > thank you again for your positive assessment of our work. We would be happy to follow up on any remaining questions during the remainder of the discussion phase.

---

### Author Response · Authors · 2025-08-08
**Rebuttal & Discussion Summary**

Dear AC,
Dear Reviewers,

Thank you for the constructive feedback and the thoughtful discussion over the past days. Following the suggestions of the reviewers, **we added new experiments (see rebuttal to zh3E), extended the discussion of related work (see rebuttal s1fj) and improved the description of EbC (see comment to s1fj)**. A comprehensive summary of changes to the manuscript is posted in a separate comment below.

Following the discussion phase, reviewers acknowledged our edits as follows:

- Reviewer 4zKK states the clarifications & additional results gives them "**greater confidence in the work**."
- Reviewer bGkb states "the authors have **carefully and satisfyingly answered the points**" made in the review.
- Reviewer s1fj acknowledges that "revisions have **improved the clarity of the paper**, and the **core contributions are now more accessible**" and that " inclusion of the **new experiment** based on a real-world neuroscience scenario [...] **adds empirical strength** and **highlights the practical relevance** of the proposed method".

We were also happy to see reviewers highlighting a large number of positive aspects and strengths:

- **Problem significance (Reviewer 4zKK)**: Symmetry discovery is recognized as important; the work is intriguing for this problem.
- **Clarity and presentation (Reviewer zh3E)**: The paper is described as very well-written with clear motivation and exposition; Figure 3 is noted for its intuitive visualization.
- **Clarity of concepts (Reviewer bGkb)**: The introduction is well-written and the concepts in Section 2 were presented with enough clarity to be understood.
- **Theory and guarantees (Reviewer zh3E)**: The theoretical results (e.g., Theorem 1) provide identifiability guarantees that are uncommon for self-supervised equivariant learning methods.
- **Simplicity and efficiency (Reviewer zh3E)**: The encoder-only contrastive pipeline is praised for being simple and efficient, avoiding generative-model complexity.
- **Completeness of approach (Reviewer 4zKK)**: The paper is viewed as hitting all the points of formulation, theory, algorithm, and basic experimental demonstration.
- **Completeness of submission (Reviewer s1fj)**: Reviewer s1fj acknowledges that "supplementary material is provided and includes both the theoretical proofs and additional experimental details."

In the next comment below, we detail the edits to the manuscript. We hope that they comprehensively address the reviewer's concerns.

---

> ### Author Response · Authors · 2025-08-08
> **Updates to the manuscript**
>
> Below we summarize how the discussion with the reviewers was incorporated into the manuscript. The additions to the text and experimental results are detailed in the respective rebuttals and official comments.
>
> **Edits by Section**
>
> - Section 1 (Introduction)
>     - **Expanded and updated related work discussion** (see rebuttal to s1fj for text); highlight difference in tasks (known vs unknown group actions) and method (encoder-only vs generative models, linear vs non-linear group representations) to better position EbC, **clarifying our contribution to the field** (4zKK, s1fj, zh3E)
> - Section 2 (Method / "Learning group structure with contrastive learning")
>     - Revised (see comment to s1fj for text); **clarified notation** for $X,X', Y,Y', g, R(g), \hat R(X, X')$; explained “pairs”; defined $S$ (negatives). (s1fj)
>     - **Clarified use of multiple group actions** across the dataset and how Eq. (2) estimates $\hat R(g)$ per action $g$. (bGkb)
>     - Moved **formal definition of diversity conditions** into the main text. (4zKK)
>     - **Corrected theorem 1 definition**, fixing typo $p_\phi = p_{f^{-1}}$; standardized symbols/notations. (s1fj)
>     - **Justified the batched-data assumption** with recent work (CARE, STL, U-NFT). (4zKK, bGkb)
>     - **Clarified** the use of the **terms** **'content' and 'style'** for equivariant and invariant representations. (bGkb)
> - Section 3 ("Experiment Setup")
>     - **Stated** that **synthetic** group **datasets are noiseless** by design **matching** the **theory**; pointed to noise experiments in the appendix. (bGkb)
>     - **Clarified which metrics require ground-truth latents**; reference metrics Acc(C, k) and Acc(G, k) as proxy metrics which don't require ground truth latents and point to existing results in appendix for empirical validation of these metrics as viable alternatives. (bGkb)
>
> - Section 4 ("Empirical Results")
>     - More **prominently highlighted** the **existing results** for $O(n)$, $SO(n)$, and $GL(n)$ with **$n>3$** are provided in the appendix. (bGkb, zh3E)
>
> - Section 5 ("Further Analysis, Ablations, and Modeling Choices")
>     - We **add a paragraph calling out the additional experiments** (see below) and pointing to the respective sections **in the appendix** (zh3E, bGkb, s1fj)
> - Appendix
>     - **Revised statement** of Theorem 1 and diversity conditions (bGkb)
>     - Clarified contributions relative to Roeder et al. (identifiability proof for jointly learning the latent space and the group actions). (4zKK)
>     - Included the **extensive related work discussion** posted in the rebuttal. (4zKK, s1fj, bGkb, zh3E)
>     - Added more detailed **discussion of potential applications** and types of datasets matching the required data structure. (zh3E, 4zKK, bGkb)
>
> **Additional experiments.** We added the following additional experiments to the appendix:
> - **Data efficiency** hyperparameter sweeps over dataset size, negatives, and samples per action. (zh3E)
> - **Noise robustness** in latent dynamics with varying levels of additive gaussian noise and varying samples per action. (bGkb, zh3E)
> - **Real-world validation** on time series data from neuroscience domain (recordings from rat hippocampus); applying EbC to achieve behavior-equivariant representations; comparison to CEBRA-Behavior. (s1fj, zh3E)
>
> **Presentation**
>
> - Fixed typos, standardized notation, and improved non-native english phrasing. (s1fj, 4zKK, zh3E, bGkb)
> - Revised Figure 2 and caption (see s1fj rebuttal for text): encoder is $\phi$, $Q$ → $\hat R$; aligned notation with main text. (s1fj, zh3E)

---

### Decision · Program_Chairs · 2025-09-17

**Decision:**

Accept (poster)

**Comment:**

The paper introduces a contrastive learning framework for group representation learning with theoretical guarantees of identifiability, supported by synthetic and 2D image experiments. Reviewers consistently highlight the clarity of the writing, strong motivation, and intuitive visuals. The reviewers noted the simplicity of the method: encoder-only with least-squares fitting, avoiding generative models, and also for achieving strong empirical results across diverse groups (SO(3), O(3), GL(3), product groups), often outperforming baselines. Theoretical contributions, especially Theorem 1, are noted as uncommon in equivariant self-supervised learning, though some reviewers find the novelty over prior work (e.g., Roeder et al.) marginal. Weaknesses center on missing details or limitations: (1) data efficiency and scaling behavior are untested, leaving unclear how performance depends on dataset size, batch size, or negative samples; (2) comparison to recent related baselines (e.g., Park et al. 2022, Keurti et al. 2023, Dehmamy et al. 2021, Yang et al. 2023, Yu et al. 2023) is incomplete; (3) the strong assumption that all samples in a batch share the same group action is unrealistic and unvalidated on real data; (4) experiments are restricted to synthetic datasets, without noisy or realistic applications.

Despite these critiques, the overall consensus is positive: the theoretical guarantees, simplicity of the framework, and strong synthetic performance outweigh the weaknesses, warranting acceptance. One reviewer had concerns about notation and clarity, but on following the responses and reading the paper, I find the critique to be outsized -- should be easily fixable. Given all these considerations, I would recommend acceptance. The authors are encouraged to incorporate various points that have come up during the discussion, and improve discussion of prior work.